# Selective activation of four quasi-equivalent C–H bonds yields N-doped graphene nanoribbons with partial corannulene motifs

Yixuan Gao[1,7], Li Huang[1,7], Yun Cao[1,7], Marcus Richter [2,7], Jing Qi[1], Qi Zheng [1], Huan Yang[1], Ji Ma [2], Xiao Chang[1], Xiaoshuai Fu[1], Carlos-Andres Palma [1], Hongliang Lu[1], Yu-Yang Zhang [1], Zhihai Cheng [3], Xiao Lin[1], Min Ouyang [4], Xinliang Feng [2,5] ✉, Shixuan Du [1,6] ✉ & Hong-Jun Gao [1,6] ✉

Selective C–H bond activation is one of the most challenging topics for organic reactions. The difficulties arise not only from the high C–H bond dissociation enthalpies but also the existence of multiple equivalent/quasi-equivalent reaction sites in organic molecules. Here, we successfully achieve the selective activation of four quasi-equivalent C–H bonds in a specially designed nitrogen-containing polycyclic hydrocarbon (N-PH). Density functional theory calculations reveal that the adsorption of N-PH on Ag(100) differentiates the activity of the four *ortho* $C(sp^3)$ atoms in the N-heterocycles into two groups, suggesting a selective dehydrogenation, which is demonstrated by sequential-annealing experiments of N-PH/Ag(100). Further annealing leads to the formation of N-doped graphene nanoribbons with partial corannulene motifs, realized by the C–H bond activation process. Our work provides a route of designing precursor molecules with *ortho* $C(sp^3)$ atom in an N-heterocycle to realize surface-induced selective dehydrogenation in quasi-equivalent sites.

Carbon-hydrogen (C–H) bond activation in a molecule is an efficient yet challenging route towards synthesis of complex organic compounds[1–5] due to the existence of multiple reactive sites[6,7]. By introducing heteroatoms, such as nitrogen and sulfur atoms into the organic skeleton, the *ortho* C atoms can be more reactive than other C atoms[8–14]. This strategy, however, lacks selectivity among equivalent/quasi-equivalent *ortho* C atoms, as previous studies show all the *ortho* C atoms have almost equal probability to be activated[8,10,12,15]. Therefore, in route towards fine control over C–H selectivity, the selective activation of equivalent/quasi-equivalent *ortho* $C(sp^3)$-H would be a major achievement in the field.

In recent years, on-surface chemistry has become a quickly developing field and provides an important route for synthesizing complex organic compounds benefiting from extraordinary controllability over the molecular formation process[1,16–26]. In addition, surface characterization techniques, such as scanning tunneling microscopy (STM) and non-contact atomic force microscopy (nc-AFM) combined with density functional theory (DFT) calculations, have been demonstrated to be an invaluable strategy to allow unambiguous characterization of the complex chemical structures and further elucidation of their underlying mechanism[27]. The surface could provide the equivalent functional groups with variable activity due to different binding

[1]Institute of Physics & University of Chinese Academy of Sciences, Chinese Academy of Sciences, Beijing 100190, PR China. [2]Center for Advancing Electronics Dresden (cfaed) & Faculty of Chemistry and Food Chemistry, Technische Universität Dresden, D-01069 Dresden, Germany. [3]Department of Physics and Beijing Key Laboratory of Optoelectronic Functional Materials & Micro-nano Devices, Renmin University of China, 100872 Beijing, China. [4]Department of Physics, University of Maryland, College Park, MD 20742, USA. [5]Max Planck Institute of Microstructure Physics, Weinberg 2, Halle 06120, Germany. [6]Songshan Lake Materials Laboratory, Dongguan, Guangdong 523808, PR China. [7]These authors contributed equally: Yixuan Gao, Li Huang, Yun Cao, Marcus Richter. ✉e-mail: xinliang.feng@tu-dresden.de; sxdu@iphy.ac.cn; hjgao@iphy.ac.cn

affinity, which has the potential to induce asymmetric active sites in symmetric molecules[28]. By carefully designing the molecular symmetry, geometric size and orientation on a substrate, it is possible to realize selective activation of C–H bond among equivalent/quasi-equivalent sites.

In this study, we designed a nitrogen-containing polycyclic hydrocarbon, namely, 10,19,21-trihydro-8H-pyrido[3,2,1-de]phenanthridine[2,3-j]isoquinolino[4,3,2-de]phenanthridine (**1**), which contains four *ortho* C(sp³) atoms on N-heterocycles and put it on a Ag(100) substrate. DFT calculations unveiled that the four *ortho* C(sp³)-H bonds in two adjacent N-heterocycles in **1** have different vibration energies on Ag(100) due to distinct adsorption sites, suggesting selective activation of *ortho* C(sp³)-H bonds. In experiment, we first deposited **1** on Ag(100) and then observed the hierarchical detachment of two pairs of *ortho* C(sp³)-H bonds at different annealing temperatures by STM and nc-AFM. The two *ortho* C(sp³)-H bonds close to the center (α sites) of **1** cleaved at 573 K, resulting in an intermediate with a quinodimethane core. With increasing temperature to 623 K, the other two *ortho* C(sp³)-H bonds (β sites) were activated and the *ortho* C(sp²) kept unchanged, resulting in the formation of polycyclic aromatic azomethine ylide (PAMY) dimer species **3** afterwards. Therefore, we successfully in-situ generate the high reactive PAMY dimer species **3** by cleaving four quasi-equivalent *ortho* C(sp³)-H bonds on Ag(100) hierarchically using **1** as precursor, in good agreement with DFT calculations. Finally, the reactive **3** on the surface enabled the formation of N-doped graphene nanoribbons through dehydrogenative intermolecular cross-coupling upon further annealing at 673 K. It is noteworthy that the *ortho* C(sp³) atoms are more active than the *ortho* C(sp²) atoms in the experiment. Further DFT calculations on two simple N-heterocycle containing molecules, pyridine and 1,2-dihydropyridine, suggest that it is a universal phenomenon in N-heterocycles and provide a strategy to differentiate the activity in quasi-equivalent sites.

## Results and disscussion

### Selective activation prediction of quasi-equivalent C–H bonds

The $C_2$ symmetric precursor (**1**), which was synthesized in three steps starting from 2,6-dibromoaniline (**7**, the detailed synthesis route is described in Supplementary Information), has four *ortho* C(sp³) atoms in N-heterocycles that can be divided into two groups, i.e., the α sites (marked in blue in Fig. 1) and the β sites (marked in red in Fig. 1). For isolated **1**, the phonon energies of C(sp³)-H stretching modes are almost the same at α and β sites (Supplementary Fig. 24), indicating the bond strength of these C(sp³)-H are similar.

In order to differentiate the bond strength of the C(sp³)-H at α and β sites, we utilized the $C_{4h}$ symmetric Ag(100) substrate for the adsorption of **1** that can keep its $C_2$ symmetry. As shown in Fig. 1a, b, the adsorption orientations of **1** are defined as the rotation angle θ between long axis of **1** and the [110] direction of Ag(100). We first performed potential energy surface (PES) calculations by scanning 45 configurations with different rotation angles, as shown in Supplementary Fig. 25. Further relaxation finds that the configuration with rotation angle θ = 22° (light coral stripe in Fig. 1b) is most stable for **1** on Ag(100).

In the most stable configuration, the adsorption sites of the C atom at α and β site locate close to hollow site and atop site of Ag(100), respectively (Fig. 1c, left panel), leading to different calculated phonon energies of 355 meV and 361 meV for the C(sp³)-H stretching at α and β sites. The phonon energies of C(sp³)-H stretching mode can be used to correlate to the bond strength[29–32], which can serve as a tentative prediction for the selective bond dissociation. It indicates the C(α)-H bond with a relatively low vibrational frequency is easier to dissociate than C(β)-H bond.

To further solidify the relative activity of C(sp³) at α and β sites, we analyze their contribution to the frontier orbitals. Based on the frontier molecular orbital theory[33], the highest occupied molecular orbital

(HOMO) and the lowest unoccupied molecular orbital (LUMO), are mainly responsible for chemical reactions[34,35]. Since the *ortho* C(sp³) atoms (C(α) and C(β)) in N-heterocycles are active sites, their contributions to the HOMO or LUMO will determine their activity. The dissociation of C–H bond involves HOMO due to the loss of H atom. Therefore, we defined a term $RA_{C(α)/C(β)}$ as the ratio of electron density of HOMO contributed from C(α) to that from C(β) (detailed information can be found in Supplementary Fig. 26) and plotted the data by the purple stars in Fig. 1b. The calculated $RA_{C(α)/C(β)}$ in each adsorption configuration is >1, indicating the C(sp³) atoms at α site are more active than β site on Ag(100) surface. It is worth mentioning that $RA_{C(α)/C(β)}$ is significantly large (almost 1.6) in the θ = 22° adsorption configuration, which means the differentiation of the activity of C(sp³) atoms at α and β sites by adsorption on Ag(100) surface is feasible.

Both of the C(sp³)-H phonon energy and the $RA_{C(α)/C(β)}$ value as discussed above suggest that C(α)-H bond may dissociate first and produce an intermediate with a quinodimethane core **2**, as shown in the middle panel in Fig. 1c. Then the C(β)-H bonds will dissociate and lead to the product PAMY dimer. PAMY dimer has four major resonance structures, the ionic **3**, the radical **3′** form, the quinoid resonance stabilized **3\*** and **3\*\***, as shown in Supplementary Fig. 28. The two Lewis-**3** and **3′** structures are energetically favored than **3\*** and **3\*\*** due to more Clar sextets (blue circles)[36]. In order to identify the most stable resonance structure of PAMY dimer on Ag(100), only the ionic **3** and the radical **3′** forms are considered here, as shown in the right panel of Fig. 1c.

To identify the actual resonance structure of the PAMY dimer on Ag(100), we first compared its C-N bond lengths with those of the isolated **3** and **3′** based on spin-polarized DFT calculations (Supplementary Table 1). The bond lengths of C(β)-N are smaller than C(α)-N in isolated **3** while it reverses in isolated **3′**. For PAMY dimer on Ag(100), the relative bond lengths of those C-N bonds are consistent with the isolated **3**. Then, we calculated the density of states (DOS) of PAMY dimer on Ag(100), which shows an energy gap (-1.0 eV) in right panel of Fig. 1d that is similar with the energy gap (0.89 eV) of isolated **3** in Fig. 1e. For comparison, the calculated energy gap of isolated **3′** is only 0.15 eV (Fig. 1f). In addition, the electron density of HOMO of PAMY dimer on Ag(100) mainly distributes at the *ortho* C atoms with little contribution from N atoms, while the electron density of LUMO is distributed over all N atoms and *ortho* C atoms. It shows similar feature to that of the isolated ionic structure **3**. Therefore, both the bond lengths and the frontier orbitals indicate that product PAMY dimer on Ag(100) prefers to stay in the ionic structure **3** rather than diradical structure **3′**.

### Possible dehydrogenation paths in the N-heterocycles

To further analyze the reaction path and the underlying mechanism of selective dehydrogenation of **1**, we calculated the dehydrogenation energy barriers of the four C(sp³)-H bonds at α and β sites using the climbing image nudged elastic band (CI-NEB) method, as shown in Fig. 2. The energy barrier for the removal of an H atom at α site is 1.05 eV, which is 0.05 eV lower than that at the β site (Fig. 2 Step 1), indicating that the initial cleavage of C(sp³)-H bond is most likely to happen at α sites, and a corresponding metastable state **2a** is yielded as shown in the lower panel of Fig. 2.

**2a** has a C(sp²) atom in radical form at one of the α sites and three C(sp³) atoms at the other α site and both β sites, which results in three possible reaction sites for the removal of the second H atom, namely H2, H3 and H4. The energy barriers for the removal of these H atoms are 0.57 eV (H2), 0.60 eV (H3), and 1.06 eV (H4), respectively, indicating a preference of removal of H2 atom at the C(sp³) atom at α site resulting in intermediate **2** (Fig. 2 Step 2, Supplementary Fig. 30). This intermediate has a closed-shell structure that contains an eight π-electron quinodimethane core. Considering the detachment of H atoms can be excited by thermal activation, ab initio molecular

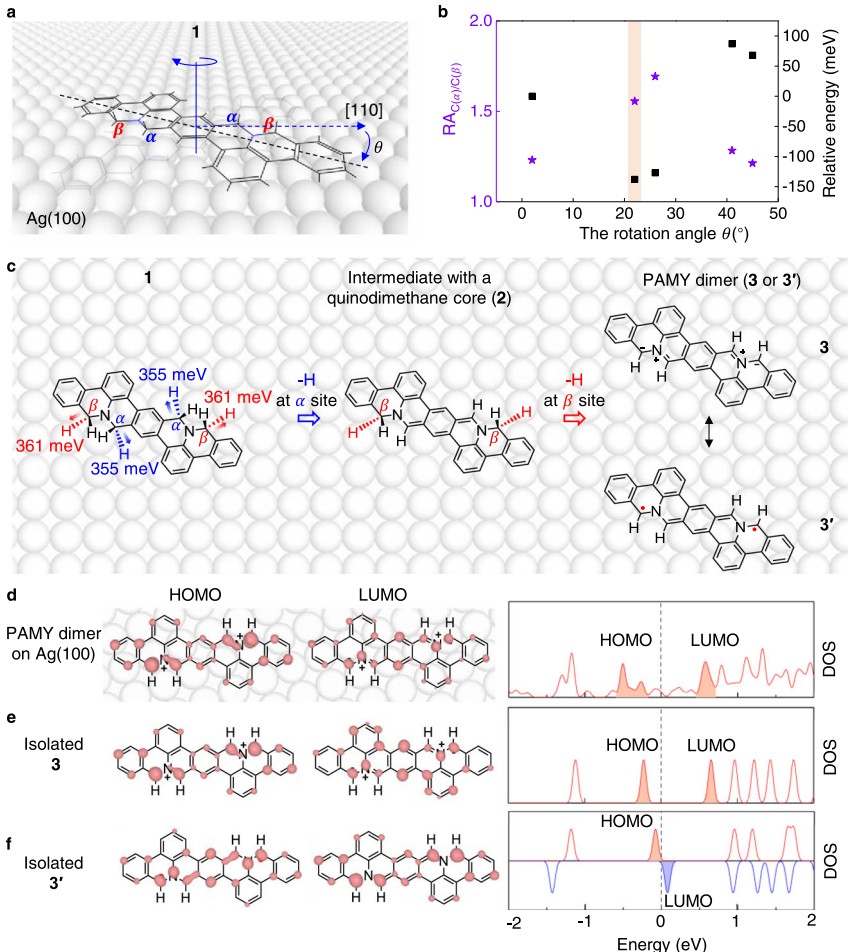

**Fig. 1 | DFT calculations of the possible adsorption configurations of 1 on Ag(100), specific C(sp³)-H vibrational modes in the most stable configuration, DOS and electron distribution of product PAMY dimer on Ag(100).**
**a** Schematics of adsorption configuration of **1** on Ag(100) with different orientations. The long axis of **1**, the [110] direction of Ag(100), and rotation axis are marked with black dash line, blue dash arrow and blue line. The rotation angle is denoted by $\theta$. **b** The comparison of relative activity of C atoms at $\alpha$ and $\beta$ sites ($RA_{C(\alpha)/C(\beta)}$, purple stars) and relative energy (black square) of each adsorption configuration with respect to the rotation angle $\theta$. Relative energy is defined as the energy difference between different adsorption configurations and the one with a rotation angle of 2 degree. The most stable configuration with rotation angle $\theta = 22°$ is highlighted by a stripe in light coral. **c** Specific C(sp³)-H vibrational modes in the most stable configuration. Left panel: two degenerate phonon modes at 355 meV represent the C(sp³)-H stretch at $\alpha$ sites (marked in blue). Two degenerate phonon modes at 361 meV represent the C(sp³)-H stretch at $\beta$ sites (marked in red). Middle and right panels: possible intermediate **2** and product PAMY dimer after detachment of H atoms from C atoms at $\alpha$ and $\beta$ sites. **d-f** The electron distribution of HOMO, LUMO and DOS of PAMY dimer on Ag(100), isolated **3** and **3'**, respectively. Spin up and spin down DOS of isolated **3'** are marked in red and blue lines, respectively. The electron distribution of each HOMO and LUMO in the left panels is the integration of electrons in the energy range filled with light coral and light blue colors shown in the right panels. The iso-surface of HOMO and LUMO of PAMY dimer on Ag(100) are 0.003 $e$/Bohr³ and 0.004 $e$/Bohr³, respectively. The iso-surface of HOMO and LUMO of isolated **3** and **3'** are 0.004 $e$/Bohr³ and 0.002 $e$/Bohr³, respectively.

dynamics (AIMD) simulation was performed to further verify the process. According to the AIMD results, the detachment of H atoms at $\alpha$ sites (H1 and H2) occurs while the H atoms at $\beta$ sites remain unchanged (Supplementary Fig. 31). This is in accordance with the sequence of dehydrogenation as predicted by the DFT-calculated energy profile.

After the detachment of the hydrogen atoms at $\alpha$ sites, intermediate **2** still has two C(sp³) atoms at $\beta$ sites. At this stage, the detachment of H3 (H3') or H4 (H4') are symmetrical equivalent. The detachment of H3 first and then H4 has the energy barriers of 1.97 eV and 0.83 eV (Fig. 2 Step 3 and 4), respectively, which results in the final product **3**. We notice the barrier for dissociation of the C(sp³)-H3 bond is much higher (1.97 eV) when starting from the metastable state **2**, compared to that starting from the initial state **1** (1.10 eV). The reason for the increased barrier is a less conjugated transition state **TS-3a** in the pathway starting from metastable state **2** (Supplementary Fig. 33). The lower panel of Fig. 2 summarizes the energy-favorable dehydrogenation path, in which the $\alpha$ site H1 atom detaches first, then $\alpha$ site

H2, followed by $\beta$ site hydrogen atoms H3 and H4. For comparison, the configurations of all the initial states, transitional states, metastable states and final states in the calculated dehydrogenation paths are shown in Supplementary Fig. 29.

The CI-NEB calculations on dehydrogenations of **1** on Ag(100) shows that the initial removal of the H1 atom has a significantly higher energy barrier (1.05 eV) than the subsequent dehydrogenation barrier of H2 atom (0.57 eV). It indicates that Step 2 can be automatically triggered after Step 1 in experiments where the temperature overcomes the 1.05 eV barrier. Thus, product **2** with simultaneous detachment of H atoms at both $\alpha$ sites should be observed. It is noteworthy that for the cleavage of the third C(sp³)-H bond in Step 3, the activation barrier increases to 1.97 eV compared with 1.05 eV for the first dehydrogenation in Step 1, which suggests higher annealing temperature required for the detachment of H atoms at $\beta$ sites to yield product **3**. Therefore, with controlled annealing experiments, hierarchical dehydrogenation at $\alpha$ sites and $\beta$ sites should be achieved in **1** on Ag(100).

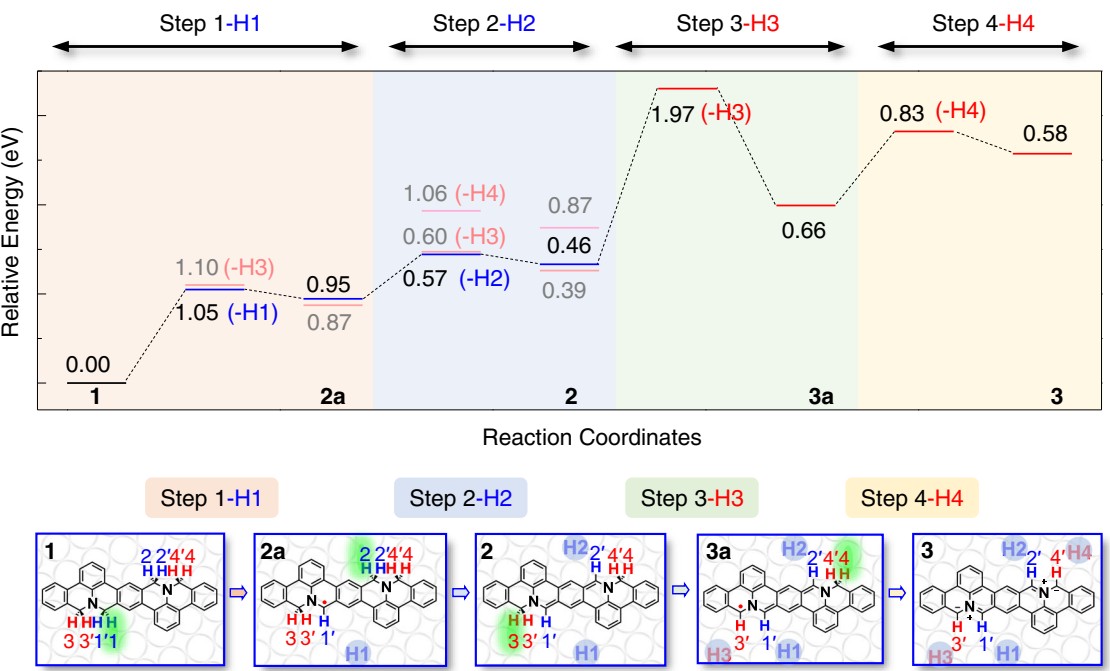

**Fig. 2 | DFT calculations on dehydrogenation barriers of the four *ortho* C(sp³) atoms in the N-heterocycles in 1 on Ag(100).** The upper panel, DFT-calculated energy profiles along all possible dehydrogenation paths. The lower panel, schematics of initial state (**1**), three metastable states (**2a**, **2**, **3a**) and final state (**3**) of dehydrogenation process we used in the calculations. In Step 1, H1 or H3 atoms detach from **1**. The dehydrogenation barrier of H1 is lower than that of H3. In Step 2, H2, H3 or H4 atoms detach from **2a**. The dehydrogenation barrier of H2 is the lowest. In Step 3 and Step 4, H3 atom detaches from **2** and H4 atom detaches from **3a**, respectively. The energy-favorable profile is highlighted while the others are in light colors. According to the energy profile, the dehydrogenation sequence is H1, H2, H3 and H4, which are highlighted by green ovals in the lower panel.

## Experimantal realization of selective activation

To verify the reaction process predicted above, sequential-annealing experiments were performed and investigated using STM and nc-AFM. When sublimated onto Ag(100) surface, **1** form chiral adsorbates as manifested in Fig. 3a. The rotation angle between the long axis of **1** (yellow dashed line in Fig. 3a) and the <110> directions of Ag(100) is $19 \pm 1°$, in good agreement with that from DFT calculations, 22°. Besides their chirality, **1** have similar features in the STM and nc-AFM images. As shown in zoomed-in images in Fig. 3b, c, **1** has slightly weakened contrast and sharp ridges distributed symmetrically around the N atoms (as marked by red dotted circles) in STM and nc-AFM image, respectively. Each bright ridge originates from two out-of-plane C(sp³)-H bonds that contribute more to the repulsive forces[37,38] at adjacent $\alpha$ and $\beta$ sites around on N atom. The atomic resolution Laplacian filtered nc-AFM image exhibits excellent consistency with the superposed molecular structure (Fig. 3d).

To activate C(sp³) atoms, the sample was first annealed at 573 K for 30 min, which resulted in molecules with brighter contrast around the N atoms than **1** in the STM image (as marked by red dotted circles in Fig. 3f). The corresponding nc-AFM image displays bright protrusions that locate at the outer side of the N dopants (red dashed circles in Fig. 3g) indicating the existence of two single out-of-plane C(sp³)-H bonds at $\beta$ sites and the cleavage of C(sp³)-H bonds at $\alpha$ sites, which gives the $\alpha$ site C atoms planar configurations. These features are consistent with the chemical structure of **2** (Fig. 3h), verifying that **1** has turned into **2** after annealing at 573 K with the activation of C(sp³) atoms at $\alpha$ sites.

Further annealing the sample at 623 K for 30 min results in a new appearance with weak contrast at *ortho* C atoms, as displayed in the zoomed-in STM image in Fig. 3j. The nc-AFM image (Fig. 3k) shows that there are no bright protrusions in the red dashed circles, which indicates that the two C(sp³) atoms at $\beta$ sites also go through the dehydrogenation process. Combined with the STS measurement which follows the calculated DOS spectral shape of **3** on Ag(100)

(Supplementary Fig. 36), we assign the product as **3** with a planar configuration. Furthermore, the regions around both N atoms appear to be darker than the neighboring C atoms, similar to the nc-AFM images on nitrogen doped graphene nanoribbons[39] and should be attributed to the different onset short-range repulsion distance between C and N atoms. The experimental observations of **2** and **3** demonstrate the successful realization of selective hierarchical C(sp³) atoms activation of the N-containing monomer **1** by sequential annealing, and confirm the DFT calculation predicted dehydrogenation sequence of C(sp³) atoms at $\alpha$ sites and $\beta$ sites.

Furthermore, we found that the detachment of H atoms only occurs on the two $\beta$ site C(sp³) atoms but not on the two $\alpha$-site C(sp²) atoms when we anneal the sample at 623 K as shown in Fig. 3. The phenomenon implies that the *ortho* C(sp³) atoms are more active than *ortho* C(sp²) atoms in the intermediate. Meanwhile, the calculated phonon energies of *ortho* C(sp³)-H stretching mode in **2** is 25 meV lower than that of *ortho* C(sp²)-H, which is consistent with the experimental observations. In order to establish the universality of the finding, we investigated two simple molecules containing N-heterocycles, pyridine and 1,2-dihydropyridine. The calculated phonon energies of *ortho* C(sp²)-H stretching mode in pyridine is higher than that in two *ortho* C(sp³)-H in 1,2-dihydropyridine both in isolated states or adsorbed on Ag(100) (Supplementary Fig. 35). It suggests that the *ortho* C(sp³) atom are more active than *ortho* C(sp²) atoms in a N-heterocycle. Therefore, introducing *ortho* C(sp³) could be an effective strategy to differentiate the activity of *ortho* C atoms in N-heterocycles.

Though the dehydrogenation happens at elevated temperatures, the experimental observations agree well with our prediction based on DFT calculations at 0 K. We therefore investigate the possible adsorption configurations of precursor **1** on Ag(100) at 573 K. Guided by the Arrhenius equation, the population ratio of the ground state to other local minima is 39: 1. Such a large ratio indicates most of the molecules stay in the ground state at the actual reaction temperature,

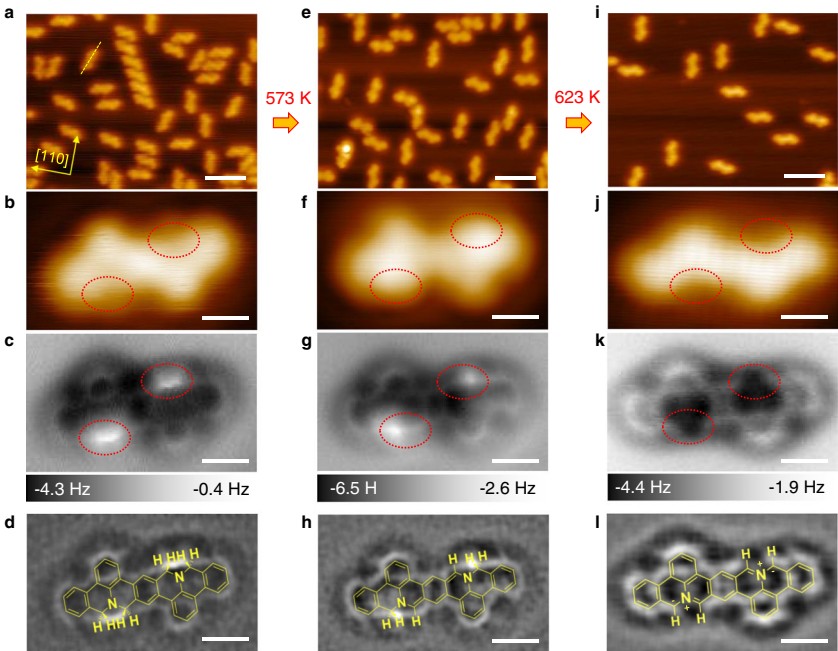

**Fig. 3 | STM and nc-AFM images of 1 upon sequential annealing on Ag(100). a–d** As-deposited **1** on the Ag(100). The long axis of a **1** monomer is marked with yellow dash line. **e–h** After annealing at 573 K for 30 min, most **1** transformed to **2. i–l** After annealing at 623 K for 30 min, most of the molecules transformed to **3. a, e, i** Large scale and **b, f, j** zoomed in STM images. **c, g, k** Zoomed in and **d, h, l** Laplacian filtered bond-resolved nc-AFM images, corresponding chemical structures are superposed in **d, h, l**. The red dotted circles in the middle panels highlight the *ortho* C–H positions. Scanning parameters: the nc-AFM images: $V_s = -30$ mV, $I_t = 40$ pA; the nc-AFM images: Amplitude = 100 pm. Scale bars: **a, e, i** 5 nm, **b–d, f–h, j–l** 5 Å.

suggesting the prediction based on DFT calculations is reasonable. More details can be found in Supplementary Fig. 25.

**The formation of N-doped GNRs with partial corannulene motifs**
To further investigate C(sp$^2$) atoms activation in **3**, we increased the annealing temperature to 673 K which provides W-shaped ribbons as shown in Fig. 4a. The intermolecular coupling could be induced by **3** of heterochirality or homochirality, among which heterochirality coupling results in uniform W-shaped ribbons (ribbon **5**; possible intermolecular couplings induced by homochirality are shown in Supplementary Fig. 37). Figure 4b presents a zoomed-in STM image of a W-shape ribbon acquired with a CO functionalized tip, in which the monomers with different chirality are denoted in red and blue contours. We analyzed 344 linkages in the ribbons in five high-resolution images, and found that 47.7% of them are hetero-chiral linkages (Supplementary Fig. 38). Due to the randomness of homo-chiral or hetero-chiral coupling, most ribbons are made of both hetero- and homo-chiral linkages, resulting in distorted shapes. Even though, there are at least two W-shaped ribbons that consist of >4 monomers in every 100 × 100 nm$^2$ area, according to our large-area STM images (Supplementary Fig. 39).

From the high-resolution nc-AFM image (Fig. 4c) of a section of **5**, the chemical structure of **5** and the corresponding reaction processes can be deduced as presented in Fig. 4f, in which **3** goes through complex chemical transformations during the polymerization processes. DFT calculations show the energy barriers for the removal of an H-atom in **3** at α site, β site and the site at the neighbor of the N-heterocycles are all higher than that for C-N cleavage (as shown in Supplementary Fig. 40), which indicates the C–N bond cleavage is favored over the C–H bond cleavage in this step. Consequently, C–N bond (highlighted in red in **3**) cleavage is induced to form the **4a**. Then, the carbene of **4a** (highlighted with yellow shadow) rotates around the single bond, producing transient **4b**. Finally, alternating cycloaddition dehydrogenation reaction between terminal carbene and *ortho* C(sp$^2$) atoms of another **4b** with different chirality generates the ribbon **5**.

Meanwhile, strained pentagons (green shadows in **5**) are generated with partial corannulene motifs, which adopt a non-planar shape in the ribbon **5**. The newly formed bonds are highlighted in red in **5** (Fig. 4f). The DFT-calculated band structure shows that the freestanding W-shaped ribbon has a narrow direct band gap of 0.16 eV (Supplementary Fig. 41).

At the polymerization temperature of 673 K, we also observed a small number of molecules **6** on surface. The corresponding STM and nc-AFM images (Fig. 4d, e) clearly show that the compound **6** is a direct reaction product from the intra-coupling of **4b**, which helps to justify the reaction process proposed in Fig. 4f.

In summary, we successfully demonstrate the selective hierarchical C–H activation in a N-doped model molecule **1** toward the PAMY dimer **3**. The *ortho* C(sp$^3$) atoms in N-heterocycles have higher reactivity than all the other C(sp$^2$) atoms in **1**, including the *ortho* C(sp$^2$) atoms, which is confirmed by both DFT calculations and sequential-annealing experiment. When deposited on Ag(100) substrate, the quasi-equivalent *ortho* C(sp$^3$) atoms at α sites and β sites in **1** have different binding affinity, which results in dehydrogenation process first at α sites and then at β sites with increasing temperatures. Moreover, N-doped graphene nanoribbons with partial corannulene motifs can be fabricated through complex chemical transformations when further annealing of PAMY dimer **3**. Our findings provide a design strategy to differentiate the activity in quasi-equivalent sites with *ortho* C(sp$^3$) atom in a N-heterocycle by choosing suitable substrates to realize selective dehydrogenation. This strategy enables the selective C–H activation and further specific bond transformation or chemical substitution.

## Methods
### Experimental methods
The experiments were carried out in a Createc low-temperature STM/AFM system in ultra-high vacuum (with a base pressure better than 3.0 × 10$^{-10}$ mbar). The measurements were conducted at 4.5 K. A qPlus sensor (Q factor = 20,000, resonant frequency = 28 kHz) with Pt-Ir tip

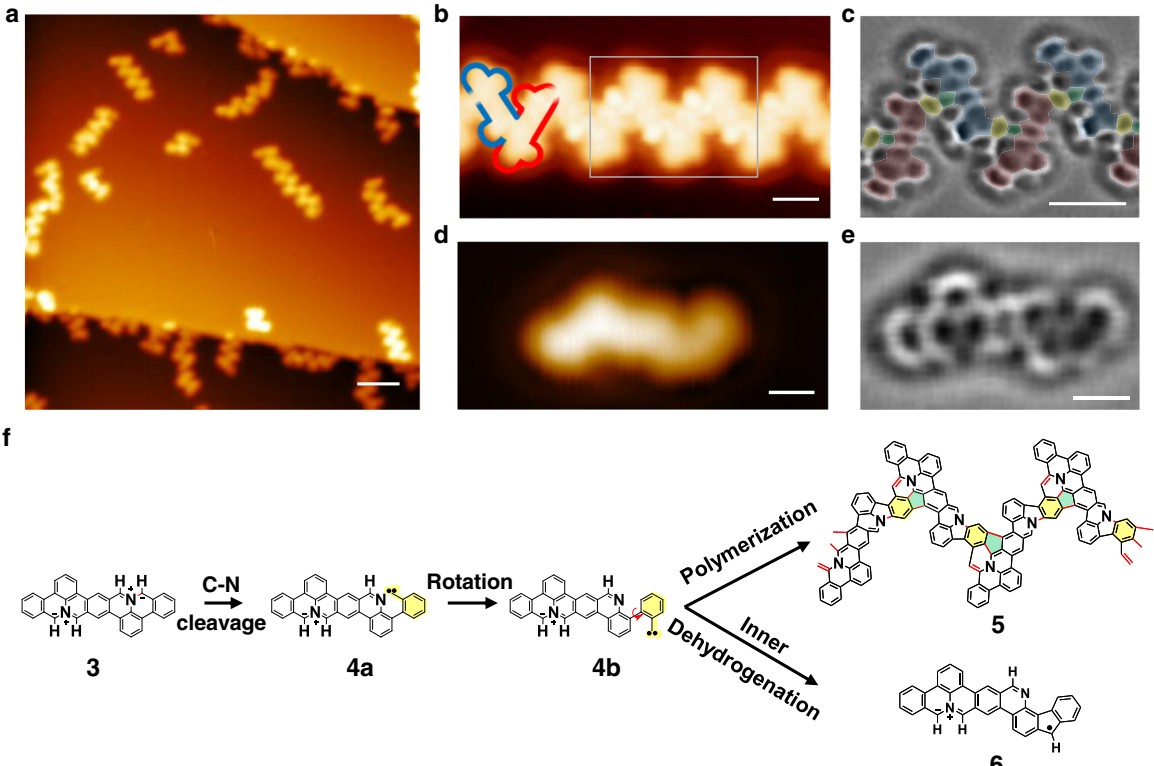

**Fig. 4 | STM and nc-AFM images of ribbons formed through 3 coupling after annealing at 673 K and the schematics of the process. a** Large-scale STM image of the ribbons. **b** STM image of a W-shaped ribbon **5** acquired with a CO functionalized tip. The **5** is constructed by enantiomers with different chirality. We highlight two enantiomers with different chirality using blue and red contours. **c** Laplacian filtered bond-resolved nc-AFM image taken in the gray box in **b**. The enantiomers with different chirality are painted in light blue and light pink; the six-membered rings of carbenes are colored in light yellow; and the newly formed pentagons are filled in light green. **d** STM image and **e** Laplacian filtered nc-AFM image of side product **6**. **f** The proposed reaction processes. The chemical structures of **5** and **6** are resolved from **c** and **e**. The carbenes of **4a** and **4b** are highlighted with yellow shadows. The red bonds and green shadows in **5** denote the newly formed bonds and pentagons in polymerization, respectively. The yellow shadows in **5** denote the six-membered rings of carbene. Scanning parameters: (**a**) $V_s = -1$ V, $I_t = 10$ pA; (**b**) $V_s = -40$ mV, $I_t = 60$ pA; **c** and **e** Amplitude $= 100$ pm; **d** $V_s = -40$ mV, $I_t = 10$ pA. Scale bars: (**a**) 5 nm; (**b**) and (**c**) 1 nm; (**d**) and (**e**) 0.5 nm.

was used for STM/nc-AFM measurements. The STM characterization was performed in constant current mode and the bias refer to the voltage on samples with respect to the tip. AFM data were taken with CO functionalized tip in constant height mode with oscillation amplitude of 100 pm. The images were processed using WSxM software package. The Ag(100) substrate was treated by repeated cycles of standard Ar⁺ sputtering and annealing at 750 K to obtain atomically flat clean surface. **1** was synthesized according to the procedures as described in SI. The sample temperature was measured with a thermocouple.

### Theoretical methods
All DFT calculations were carried out using Vienna ab-initio simulation package (VASP)[40] with the projector augmented wave (PAW) method[41,42]. The energy cut-off of the plane-wave basis sets was 400 eV, and K-point sampling was done only at the Γ point. Van der Waals (vdW) interactions were considered at the vdW-DF level[43], with the optB86 functional was used for the exchange potential[44,45]. A slab model was used with three Ag layers as the substrate. In all the calculations, a 1.2 nm vacuum layer was used, and all atoms except the bottom Ag layer were fully relaxed until the net force was <0.1 eV/nm. Transition states were calculated using a combination of the climbing image nudged elastic band (CI-NEB). The closed-shell configuration was calculated by performing unrestricted spin-polarized energy minimization or spin-unpolarized calculations. The open-shell configuration was considered by keeping a spin multiplicity of 3 for the total spin of the molecule during the spin-polarized total energy minimization. Specific hybrid functional[46] was applied for the calculations

of the density of states of diradical state presented in Fig. 1f. For ab initio MD simulations, a canonical (NVT) ensemble was used at 1400 K, 1600 K and 2400 K. The high temperature is used to facilitate the transition process. The time interval between each step is 1 fs.

### Reporting summary
Further information on research design is available in the Nature Research Reporting Summary linked to this article.

### Data availability
The authors declare that the data supporting the findings of this study are available within the paper and its Supplementary Information files. The data that support the findings of this study are available from the corresponding authors upon request.

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

## Acknowledgements

We acknowledge the financial support from the National Natural Science Foundation of China (No. 61888102, 61622116 and 11974403), National Key Research and Development Program of China (No. 2018YFA0305800 and 2019YFA0308500), the Strategic Priority Research Program of Chinese Academy of Sciences (No. XDB30000000), the German Research Foundation (DFG) with EnhanceNano (No. 391979941), the CAS Project for Young Scientists in Basic Research (YSBR-003), the CAS Frontier Sciences and Education grant (No. QYZDBSSW-SLH038), the K. C. Wong Education Foundation and the CAS Pioneer Hundred Talents Program. Z.H. Cheng was supported by the Fundamental Research Funds for the Central Universities and the Research Funds of Renmin University of China (No. 18XNLG01). Part of the research was performed in the Key Laboratory of Vacuum Physics, Chinese Academy of Sciences. Computational resources were provided by the National Supercomputing Center in Tianjin Municipality, China.

## Author contributions

H.-J.G., S.D., and X.F. supervised the work; H.-J.G., S.D., C.-A.P. and X.F. conceived the project and designed the studies; Y.G. Y.-Y.Z. and S.D.

carried out theoretical calculations; L.H., Y.C., J.Q., Q.Z., H.Y., X.C., X.F., C.-A.P., H.L., Z.C., and X.L. performed STM/AFM experiments; M.R., J.M. and X.F. synthesized the molecules; M.O. contributed to the discussion of possible chemical reactions. The manuscript was written with contributions from all authors. Y.G., L.H., Y.C. and M.R. contributed equally to this work.

## Competing interests

The authors declare no competing interests.
