## [Peer Review File · Nature Communications]

Selective activation of four quasi-equivalent C-H bonds yields
N-doped graphene nanoribbons with partial corannulene
motifsREVIEWER COMMENTS

Reviewer #1 (Remarks to the Author):

This manuscript reports stepwise C-H bond activations of a nitrogen-substituted polycyclic compound on Ag(100) surface in UHV conditions by means of STM and nc-AFM. Moreover, DFT simulations support the experimental observations. Overall, the reported findings are of interest to the scientists in the broad community. The quality of the surface observation is high enough. Thus the reviewer recommends to accept this manuscript if the authors consider the comments listed below.

- (1) Please insert page numbers. It is difficult to indicate the sentence which I would like to comment on.
- (2) On page 5, 1st line. The authors mentioned PAMY dimer has two resonance structures. However, the compound has other resonance structures.
- (3) In Supporting Information, Figures S4, S5, S6, S7, S8, S9, S10, S12, S13, S14, S15, and S18, the NMR spectra indicate that the samples contain impurities. The reviewer strongly recommends to purify these samples. The reviewer understands that the main conclusion of the manuscript is not affected by the impurity at least at a single molecular level. Moreover, the impurity might not be sublimated. However, if there are impurities on the surface, these influence the final polymerization process. Moreover, from a general, organic chemistry standard, these impurities are not acceptable.
- (4) Please include the DFT simulations for the C-N bond cleavage process. Please discuss why the C-N bond cleavage is favored over the C-H bond cleavage.

Reviewer #2 (Remarks to the Author):

Y.X. Gao and coworkers report on the selective on-surface C-H bond activation at sp³ carbon in an N-heterocyclic molecule on Ag(100), using STM, AFM and DFT calculations. The sequential selective C-H bond dissociation is convincingly demonstrated in theory and experiment, while the second part (formation of nanoribbons) requires additional data. This manuscript may be recommended for publication in Nature Communications after the authors have addressed the issues listed below. Unfortunately, the manuscript has neither page numbers nor line numbers, so I will cite the lines to which my comments refer, if necessary.

1. The discussion of the previous work could be more specific to on-surface C-H bond activation using a more appropriate choice of literature, especially as there are only few examples of ortho-selective C-H bond activation in the context of on-surface synthesis.
2. The authors calculated several adsorption configurations, which differ in their relative energy and their catalytic activity regarding selective C-H bond activation. The critical parameter is the angle between the molecule's long axis and a high-symmetry direction. This raises the following questions: First, can the authors exclude that there are even lower energies at other angles (e.g. between zero and 22 degrees)? Second, is the energy difference between the different configurations large enough to keep the molecule in the most favorable configuration at the actual reaction temperature? Or will there be an essentially equal distribution over all possible sites at this temperature, resulting in an averaged reactivity? My numerical estimates say that there are indeed rather significant differences between the populations of the different sites even at 573 K, but this important point should be discussed by the authors in the paper.
3. "Due to the close relation between the dissociation of the C-H bond and HOMO states, $RAC(\alpha)/C(\beta)$ is defined as the ratio of number of electrons of C(sp³) atom at α site and β site ranging from HOMO to Fermi level." – It remains unclear what the authors want to say here. It should be explained in detail and with references what exactly is this "close relationship between dissociation of the C-H bond and HOMO states" and what is meant by "number of electrons of C(sp³) atom ... ranging from HOMO to Fermi level".
4. "This result indicates that if a particular energy, for example 355 meV, is applied to activate C(sp³)-H bonds, only the C(sp³)-H bond at α site will dissociate. As the phonon modes of an adsorbed molecule can be excited by thermal treatment, we deduce that the cleavage of C(sp³)-H bonds at α sites should be at lower temperature than those at β sites..." - This is a questionable concept. The authors need to compare the C-H bond dissociation energies, i.e., the depths of the potential energy

minima, and not the vibration frequencies, which correspond to the second derivatives at the potential minima. Even though there are sometimes limited correlations between these two quantities (i.e., dissociation energies and frequencies), it is difficult to understand why the vague argument with the frequencies is made here when the bond energies could be obtained from the same calculations. The calculated barriers shown in Figure 2 are much more useful in this context. Regarding the statement "the phonon modes on an adsorbed molecule can be excited by thermal treatment", the authors should also keep in mind that a thermal energy of 355 meV corresponds to a very high temperature of >4000 K. So, a thermal excitation of this vibration is certainly not the decisive factor for bond dissociation here, especially as multiple excitations would be necessary to induce thermal dissociation. What the authors observe here is first and foremost a surface-induced catalytic reaction and not a thermal cracking of C-H bonds due to excessive vibrational excitation.

5. Why is the barrier for dissociation of the C-H bond H3 much higher (1.97 eV) when starting from the metastable state 2, compared to the barrier for dissociation of the same bond when starting from the initial state 1 (1.10 eV)?

6. Nanoribbons: The only large-scale SPM image (Figure 4a) shows just one ribbons with several repeat units resembling the zoom-in image in Figure 4b. All other products in the large-scale image show partially distorted shapes suggesting that they have different structures. The authors should provide statistical information regarding the yield of the W-shaped ribbons based on additional large-scale SPM data (which should be displayed in the SI).

Reviewer #3 (Remarks to the Author):

The manuscript by Gao et al. describes combined theoretical and experimental research on the hierarchical C-H bond activation in molecules synthesized from precursors directly on the Ag substrate. The investigation falls into the recently developing area of the on-surface synthesis approach. I believe that the main target of the manuscript is appealing for the broad audience. In general the research is clearly described. Before the manuscript can be accepted for publication there are, however, a few issues that the authors should clarify. I will briefly describe them below:

- the authors base their identification of the final structure on Ag (differentiation between 3 and 3') on the DFT calculations, i.e. the bond lengths and DOS. Have the authors tried to support the calculations by the experimental scanning tunneling spectroscopy measurements? The combination of theoretical and experimental reasoning would make the attribution of the actual structure much more reliable
- in the description of the dehydrogenation path the energy differences for the two first steps between the lowest and second lowest possible paths are very small, in the first step the difference between the extraction of H1 and H3 is only 0.05 eV and in the second between the H2 and H3 is only 0.03 eV. However, in the sequential description the authors describe possible paths starting only from the lowest process in the previous step. For instance one could consider extraction of H3 in the first step and then the other process (which is not considered in the description). Although the first process would be higher in energy (H3 versus H1), the second one could be lower. I understand that the path described in Figure 2 is consistent with experimental findings, but my concern is whether the calculations describe the reality precisely enough, especially taking into account the relatively small differences in the two first steps.
- the authors say that the DFT calculation predicts low band gap of the ribbons, have the authors attempted the STS measurements to extract the properties from the experiment?
- minor points: I strongly recommend that the authors read carefully the text as there are several minor language errors and sentences difficult to follow. For instance already the sentence "Experimentally depositing N-PH..." in the abstract may be misleading to suggest that the hierarchical dehydrogenation has been achieved through NC-AFM. In the manuscript there are more weird sentences; the red ovals in Figure 3 are barely visible;

In summary, I find the manuscript interesting and appealing for the broad audience and believe that it could be accepted for publication once the above mentioned issues are clarified by the authors.

Reviewer #4 (Remarks to the Author):

The authors herein present the study of the stepwise thermal dehydrogenation of a polycyclic heteroaromatic hydrocarbon on a Ag (100) surface. At the center of this work lies the observation of a hierarchical temperature-controlled dehydrogenation that can be correlated with the subtle differences in the C-H BDE or vibrational ZPE of individual C-H bonds upon adsorption on the surface. The authors highlight this work as an extension of the field of C-H activation. This is clearly done in an effort to raise the impact of their work but appears as a significant stretch. C-H activation usually implies the chemical substitution of C-H bond by a higher value C-X bond not just the cleavage of a hydrogen atom. The more appropriate analogy may be found in the conventional radical mediated oxidation reactions. Barred these comments the authors should clarify the key advancements to the field and timeliness that would justify publication in Nature Communications. It remains unclear how the observed selectivity could be used as a design strategy barred the not insignificant computational effort that was necessary to disentangle the proposed mechanisms. The reliable calculation of the preferred adsorption geometry of an arbitrary molecule barred any experimental data is an enormous challenge in itself. Adding on top of that truly subtle differences in the ZPE of 6 meV that apparently gives rise to the observed selectivity is far from a practical predictive model and well within the error of DFT calculations.

Beyond the general comments above the reviewers are left with two key questions that need to be resolved before this manuscript could be reconsidered.

1) The authors introduce RAC(α)/C(β) as a the “relative activity” of C(sp³)-H bonds flanking the N-atoms. The validity and the origin of this reference has to be discussed in the manuscript. The explanation provided on page 5 is more than just unclear and leaves the reader with an incomplete understanding of the context of this comparison.

2) Based on the extended data provided as part of the supporting information it appears like the C-H bond broken as part of the dehydrogenation steps is actually the C-H bond pointing towards the surface. The authors however focus almost exclusively on the discussion of the ZPE of the C-H bonds pointing away from the surface (see Figure 1). This is bizarre as the discussed C-H bonds remain intact in the product and are therefore “irrelevant” (beyond a secondary effect) to the selectivity. Should the discussion not focus on the actual C-H bond that is broken or are the authors suggestion a mechanism where the H-atom is removed without assistance from the underlying Ag(100) substrate? There is plenty of evidence in the literature that suggest that all dehydrogenations on coinage metal surfaces rely on the transfer of a H-atom to the surface (geometrically only possible if the C-H bond points into the substrate).

Minor corrections:

Page 3 “...surface characterizing...” change to “...surface characterization...”

Page 6 “...density of LUMO distributes at all...” change to “... density of LUMO is distributed over all...”

Page 6 “...inclines...” change to “...prefers...”

Response to Reviewers

We thank the reviewers for the constructive comments and suggestions. We have addressed all these comments point-by-point and revised the manuscript accordingly.

In this response letter, comments from the reviewers are included in *black italic* typeface. Our responses are in regular black typeface. The changes to the text in the revised manuscript are described or quoted in blue.

Response to Reviewer 1:

This manuscript reports stepwise C-H bond activations of a nitrogen-substituted polycyclic compound on Ag(100) surface in UHV conditions by means of STM and nc-AFM. Moreover, DFT simulations support the experimental observations. Overall, the reported findings are of interest to the scientists in the broad community. The quality of the surface observation is high enough. Thus the reviewer recommends to accept this manuscript if the authors consider the comments listed below.

We thank the Reviewer for his or her time and helpful comments. We appreciate the recognition that C-H bond activation and our exhaustive surface studies is “*of interest to the scientists in the broad scientific community*”. Please find below the responses to the comments point by point.

Comment 1: *Please insert page numbers. It is difficult to indicate the sentence which I would like to comment on.*

Response 1: We are sorry for the inconvenience. We have added the page numbers and line numbers in the new version.

Comment 2: *On page 5, 1st line. The authors mentioned PAMY dimer has two resonance structures. However, the compound has other resonance structures.*

Response 2: We thank the reviewer for the constructive comment. Overall, there are four major different resonance structures of the PAMY dimer (see Figure R1). With five Clar sextets each (blue circles), the two Lewis-**3** and **3'** structures are energetically favored (Springer: 1983; pp 49-58). However, in both the ionic **3** and the radical **3'** forms, there is the possibility of a quinoid resonance stabilization in **3*** and **3****. For the price of a Clar sextet, two of the *para*-standing methylene bridges can be stabilized in

a *p*-quinodimethane (pink). For both forms **3*** and **3****, the delocalized π -system is enlarged and noticeably changed in this process.

Figure R1 (Figure S28). Resonance structures of the PAMY dimer.

We have revised the manuscript in page 6 line 14 and implemented Figure S28 in the Supplementary Information.

"PAMY dimer has four major resonance structures, the ionic **3**, the radical **3'** form, the quinoid resonance stabilized **3*** and **3****, as shown in Figure S28. The two Lewis-**3** and **3'** structures are energetically favoured than **3*** and **3**** due to more Clar sextets (blue circles). In order to identify the most stable resonance structure of PAMY dimer on Ag(100), only the ionic **3** and the radical **3'** forms are considered here, as shown in the right panel of Figure 1c."

Comment 3: In Supporting Information, Figures S4, S5, S6, S7, S8, S9, S10, S12, S13, S14, S15, and S18, the NMR spectra indicate that the samples contain impurities. The reviewer strongly recommends to purify these samples. The reviewer understands that the main conclusion of the manuscript is not affected by the impurity at least at a single molecular level. Moreover, the impurity might not be sublimated. However, if there are impurities on the surface, these influence the final polymerization process. Moreover, from a general, organic chemistry standard, these impurities are not acceptable.

Response 3: We thank the reviewer for this comment. We agree with the reviewer that the NMR spectra indicate impurities. However, we would like to mention that the purification of these compounds is very challenging due to the high polarity of the OH- and NH₂-substituents.

For intermediate **9**, we performed the recycle gel permeation chromatography (re-GPC) by using chloroform as eluent for further purification. The minor impurities have been removed, as shown in the ¹H- (Figure R2) and ¹³C-NMR spectra (Figure R3).

Figure R2 (Figure S4). ^1H -NMR spectrum (300 MHz) of **9** at 298 K in DMSO-d_6 .

Figure R3 (Figure S5). ^{13}C -NMR spectrum (300 MHz) of **9** at 298 K in DMSO-d_6 .

However, the intermediate **11** with four OH-groups and two NH_2 -groups cannot be purified by standard flash column chromatography. Even by rinsing with strong polar solvents such as methanol, the intermediate **11** heavily sticks to the silica gel. Since flash chromatography fails, we investigated different purification strategies (e.g. re-GPC purification is not possible due to the poor solubility in chloroform) and concluded that the precipitation of intermediate **11** in toluene is the best way for purification. Moreover, hydrogen bonds between NH_2 - and OH-substituents in **11** prevent the free rotation of phenyl substituents. This leads to isomer formation and explains the complex

^1H - and ^{13}C -spectra at room temperature.

For precursor **1**, we observed a high instability of the precursor in the iminium species. Even short-term workup of precursor **1** under ambient conditions resulted in partial oxidation of the iminium species. Due to the high instability, we transferred the reaction mixture directly from the microwave to the glovebox. In the glovebox, we were able to purify precursor **1** by methanol precipitation.

As already mentioned by the reviewer, the impurities fortunately do not impact the surface synthesis. Moreover, no further compounds could be visualized after the evaporation of the precursor via STM (see Figure R4). Organic impurities with a small molecular weight evaporated during the heating of the crucible and subsequently decomposed. Organic impurities with a higher molecular weight than precursor **1** could not be evaporated and remained in the crucible.

Figure R4. Overview STM image after the evaporation of the precursor **1** on Ag(100) at 300 K. This figure indicates there are no other impurities on the surface. The scale bar is 5 nm.

We have added Figure R2 and R3 as Figure S4 and S5 in the Supplementary Information.

Comment 4: Please include the DFT simulations for the C-N bond cleavage process. Please discuss why the C-N bond cleavage is favored over the C-H bond cleavage.

Response 4: We thank the reviewer for the constructive suggestion. Following the suggestion, we compared the cleavage energy barriers of the C-N and C-H bonds in **3**, as shown in Figure R5 (Figure S40). The energy barriers for the removal of an H-atom at α site (Figure R5(e), β site (Figure R5(f) and the site at the neighbor of the N-heterocycles (Figure R5(b)) are 2.11 eV, 2.39 eV, and 2.41 eV, respectively, which are

all higher than that for C-N cleavage (2.02 eV). It indicates the C-N bond cleavage is favored over the C-H bond cleavage in **3**.

We have added the following discussion in the manuscript (page 15 line 11) and Figure S40 in the Supplementary Information.

"DFT calculations show the energy barriers for the removal of an H-atom in **3** at α site, β site and the site at the neighbor of the N-heterocycles are all higher than that for C-N cleavage (as shown in Figure S40), which indicates the C-N bond cleavage is favored over the C-H bond cleavage in this step. Consequently, C-N bond (highlighted in red in **3**) cleavage is induced to form the **4a**."

Figure R5 (Figure S40). DFT calculations on cleavage energy barriers of C-N and C-H bonds in **3 on Ag(100).** (a-b, e-f) Atomic structures of initial states, transition states and final states in C-N and C-H cleavage processes. (c-d, g-h) DFT-calculated energy profiles along four

paths. The white, black and blue balls represent Ag, carbon and nitrogen atoms, respectively. The detached hydrogen atom is colored red. Other hydrogen atoms are colored in light pink.

Response to Reviewer 2:

Y.X. Gao and coworkers report on the selective on-surface C-H bond activation at sp^3 carbon in an N-heterocyclic molecule on Ag(100), using STM, AFM and DFT calculations. The sequential selective C-H bond dissociation is convincingly demonstrated in theory and experiment, while the second part (formation of nanoribbons) requires additional data. This manuscript may be recommended for publication in Nature Communications after the authors have addressed the issues listed below. Unfortunately, the manuscript has neither page numbers nor line numbers, so I will cite the lines to which my comments refer, if necessary.

We thank the Reviewer for his or her time and helpful comments. Please find below the responses to the comments. We have added the page numbers and line numbers in the new version for convenience.

Comment 1: *The discussion of the previous work could be more specific to on-surface C-H bond activation using a more appropriate choice of literature, especially as there are only few examples of ortho-selective C-H bond activation in the context of on-surface synthesis.*

Response 1: We have added references related to on-surface C-H bond activation (*Science* 334, 213 (2011), *J. Am. Chem. Soc.* 141, 3550 (2019), *J. Phys. Chem. Lett.* 11, 9850 (2020)), especially the *ortho*-selective C-H bond activation (*ACS Nano* 13, 1385 (2019), *J. Am. Chem. Soc.* 138, 2809 (2016)) in the first paragraph of introduction.

Comment 2-1: *The authors calculated several adsorption configurations, which differ in their relative energy and their catalytic activity regarding selective C-H bond activation. The critical parameter is the angle between the molecule's long axis and a high-symmetry direction. This raises the following questions: First, can the authors exclude that there are even lower energies at other angles (e.g. between zero and 22 degrees)?*

Response 2-1: We thank the reviewer for the insightful comment. We followed the suggestions and performed potential energy surface (PES) calculations by scanning different rotation angles. The PES scan at different rotation angles was plotted, as shown in Figure R6, which helps to find the most stable adsorption structure at the energy minimum. We took 45 possible configurations with each rotation angle differs by 2 degree. In each configuration, the top two layers of substrate were fully relaxed, while the molecule was relaxed in z direction with a fixed in-plane orientation. Then PES is acquired after performing single point energy calculation at each rotation angle. The calculated PES shows that the configurations with rotation angles of 22 and 24 degree exhibit the lowest energies with a difference of 2 meV, as shown in Figure R6b. After further relaxation of the configuration with a rotation angle of 24 degree, it relaxes

to the configuration with a rotation angle of 22 degree. Therefore, we exclude that there are even lower energies at other angles.

Figure R6 (Figure S25). DFT Calculated potential energy surface at different rotation angles. (a) Schematics of adsorption configuration of **1** on Ag(100) with different orientations. The long axis of **1**, the [110] direction of Ag(100), and the rotation axis are marked with black dash line, blue dash arrow and blue line. The rotation angle is denoted by θ . (b) DFT calculated potential energy surface at different rotation angles θ . We took 45 possible configurations with each rotation angle differs by 2 degree. In each configuration, the molecule is fully relaxed in z direction. The configurations with rotation angles of 22 and 24 degree exhibit the lowest energies with a difference of 2 meV. Further relaxation finds that the configuration with a rotation angle of 24 degree changes to that with a rotation angle of 22 degree, which is the most possible configuration of molecule **1** on Ag(100).

We have added Figure R6 as Figure S25 in Supplementary Information and related discussion in the manuscript (page 5 line 12).

"We first performed potential energy surface (PES) calculations by scanning 45 configurations with different rotation angles, as shown in Figure S25. Further relaxation finds that the configuration with rotation angle $\theta = 22^\circ$ (light coral stripe in Figure 1b) is most stable for **1** on Ag(100)."

Comment 2-2: *Second, is the energy difference between the different configurations large enough to keep the molecule in the most favorable configuration at the actual reaction temperature? Or will there be an essentially equal distribution over all possible sites at this temperature, resulting in an averaged reactivity? My numerical estimates say that there are indeed rather significant differences between the populations of the different sites even at 573 K, but this important point should be discussed by the authors in the paper.*

Response 2-2: We agree that it is possible that the molecules may diffuse away from the most favorable adsorption configuration at actual reaction temperature of 573 K. Following the suggestion, we analyzed populations of the four configurations which are the local minima in Figure R6(b) with rotation angles of 0, 22, 52 and 66 degree at 573 K. Guided by the Arrhenius equation, which gives the dependence of the rate constant of a chemical reaction on the absolute temperature as:

$$k = Ae^{-E_{\alpha}/RT},$$

where k is the rate constant, T is the absolute temperature, A is the pre-exponential factor (A is a temperature-independent constant for each chemical reaction), E_{α} is the activation energy for the reaction, and R is the universal gas constant. At 573 K, the diffusion of the molecule can be divided into two groups. In group I, the molecules diffuse from the other local minima (configurations with rotation angles of 0, 52 and 66 degree) to the ground state (the one with a rotation angle of 22 degree). In group II, the molecules diffuse from the ground state to the other local minima. We define the rate constant of the diffusion from angle 0 degree (52 degree, 66 degree) to the ground state to be k_1 (k_1' , k_1''), and the diffusion from the ground state to the configuration with rotation angle of 0 degree (52 degree, 66 degree) to be k_2 (k_2' , k_2''). Here, we use the energy barrier as the activation energy E_{α} , as marked in Figure R6.

Thus, when $T = 573$ K,

$$k_1 = Ae^{-E_{\alpha 1}/RT} = 0.07 A, \quad k_1' = Ae^{-E_{\alpha 1}'/RT} = 0.44 A, \quad k_1'' = Ae^{-E_{\alpha 1}''/RT} = 0.24 A$$

$$k_2 = Ae^{-E_{\alpha 2}/RT} = 2.96 \times 10^{-3} A, \quad k_2' = k_2'' = Ae^{-E_{\alpha 2}'/RT} = 8.20 \times 10^{-3} A$$

The, the population ratio of the ground state to the other local minimum at 573 K is $(k_1 + k_1' + k_1'') : (k_2 + k_2' + k_2'') = 39 : 1$. Therefore, it is possible that other minima are populated and that indeed lead to 'rare catalytic events'. However, our work mainly focuses on the 'dominant catalytic events' to interpret our results in terms of molecules reacted at 573 K.

We have added the following discussion in the manuscript and Supplementary Information.

In manuscript (page 13 line 13): " Though the dehydrogenation happens at elevated

temperatures, the experimental observations agree well with our prediction based on DFT calculations at 0 K. We therefore investigate the possible adsorption configurations of precursor **1** on Ag(100) at 573 K. Guided by the Arrhenius equation, the population ratio of the ground state to other local minima is 39 : 1. Such a large ratio indicates most of the molecules stay in the ground state at the actual reaction temperature, suggesting the prediction based on DFT calculations is reasonable. More details can be found in Supplementary Information Figure S25."

In Supplementary Information: "At 573 K, the diffusion of the molecule can also be activated. The diffusion of the molecule can be divided into two groups. In group I, the molecules diffuse from the other local minima (configurations with rotation angles of 0, 52 and 66 degree) to the ground state (the one with a rotation angle of 22 degree), as shown in Figure S25; In group II, the molecules diffuse from the ground state to the other local minima. We define the rate constant of the diffusion from angle 0 degree (52 degree, 66 degree) to the ground state to be k_1 (k_1' , k_1''), and the diffusion from the ground state to the configuration with rotation angle of 0 degree (52 degree, 66 degree) to be k_2 (k_2' , k_2''). Here, we use the energy barrier to replace the activation energy E_α , as marked in Figure S25. Thus, when $T = 573$ K,

$$k_1 = Ae^{-E_{\alpha 1}/RT} = 0.07 \text{ A} , \quad k_1' = Ae^{-E_{\alpha 1}'/RT} = 0.44 \text{ A} , \quad k_1'' = Ae^{-E_{\alpha 1}''/RT} = 0.24 \text{ A}$$

$$k_2 = Ae^{-E_{\alpha 2}/RT} = 2.96 \times 10^{-3} \text{ A}, \quad k_2' = k_2'' = Ae^{-E_{\alpha 2}'/RT} = 8.20 \times 10^{-3} \text{ A}$$

Thus, the population ratio of the ground state to the other local minimum at 573 K is $(k_1 + k_1' + k_1'') : (k_2 + k_2' + k_2'') = 39 : 1$."

Comment 3: "Due to the close relation between the dissociation of the C-H bond and HOMO states, $RAC(\alpha)/C(\beta)$ is defined as the ratio of number of electrons of $C(sp^3)$ atom at a site and β site ranging from HOMO to Fermi level." – It remains unclear what the authors want to say here. It should be explained in detail and with references what exactly is this "close relationship between dissociation of the C-H bond and HOMO states" and what is meant by "number of electrons of $C(sp^3)$ atom ... ranging from HOMO to Fermi level".

Response 3: We thank the reviewer for suggesting that the arguments behind our conclusion should be clarified. Our claim of "close relationship between dissociation of the C-H bond and HOMO states" is based on the frontier molecular orbital (FMO)

theory (*J. Chem. Phys.* 20, 722-725 (1952)), which states that the frontier orbitals, i.e. the HOMO and the LUMO, are mainly responsible for chemical reactions. A recent work (*Communications Chemistry* 2, 31 (2019)) shows that the occupied C-H orbital would be expected to interact with an unoccupied *d* orbital of Ni to activate the corresponding C-H bond, in agreement with Fukui's FMO theory. Moreover, a recent work uses the differences in the electron density of singly occupied molecular orbital at the reactive position to compare the reactivity difference. (*Nature Communications* 10, 2706 (2019)).

Since the *ortho* C(*sp*³) atoms (C(α) and C(β)) in N-heterocycles are active sites, their contribution to the HOMO or LUMO will determine their activity. The dissociation of C-H bond will involve HOMO due to the loss of H atom. Therefore, in order to reveal the relationship between the dissociation of C-H bond and the frontier orbitals, we investigate the contribution of these *ortho* C(*sp*³) atoms to the HOMO state through DOS on these atoms (PDOS). Due to the split HOMO peaks after the molecule is adsorbed on substrate, we calculate the contribution of C(α) or C(β) by integrating the PDOS from the original HOMO to the Fermi level as shown in Figure R7 marked with a light coral background. The $RA_{C(\alpha)/C(\beta)}$ is then defined as the ratio of contribution from C(α) to the HOMO and that from C(β). For $RA_{C(\alpha)/C(\beta)}$ larger than 1, the C(α) is more active than C(β), and vice versa.

Figure R7 (Figure S26). (a) Projected density of states (PDOS) on C atoms at α site (upper panel) and β site (lower panel) of **TPPIP** on Ag(100). (b) The schematic of molecule **1** on Ag(100).

We have now added the following discussion in the manuscript (page 5 line 24) to

explain the close relationship between dissociation of C-H and HOMO.

"To further solidify the relative activity of C(sp³) at α and β sites, we analyze their contribution to the frontier orbitals. Based on the frontier molecular orbital theory (*J. Chem. Phys.* 20, 722-725 (1952)), the highest occupied molecular orbital (HOMO) and the lowest unoccupied molecular orbital (LUMO), are mainly responsible for chemical reactions (*Communications Chemistry* 2, 31 (2019); *Nature Communications* 10, 2706 (2019)). Since the *ortho* C(sp³) atoms (C(α) and C(β)) in N-heterocycles are active sites, their contributions to the HOMO or LUMO will determine their activity. The dissociation of C-H bond involves HOMO due to the loss of H atom. Therefore, we defined a term $RA_{C(\alpha)/C(\beta)}$ as the ratio of electron density of HOMO contributed from C(α) to that from C(β) (detailed information can be found in Supplementary Information S26) and plotted the data by the purple stars in Figure 1b. "

We added Figure R7 as Figure S26 and description of $RA_{C(\alpha)/C(\beta)}$ in the Supplementary information.

"The $RA_{C(\alpha)/C(\beta)}$ is defined as the ratio of electron density of HOMO contributed from C(α) to that from C(β). Due to the split HOMO peaks after the molecule is adsorbed on substrate (Figure S26 (a)), we obtain the contribution of C(α) or C(β) by integrating the PDOS from the original HOMO to the Fermi level as shown in Figure S26 marked with a light coral background."

Comment 4: *"This result indicates that if a particular energy, for example 355 meV, is applied to activate C(sp³)-H bonds, only the C(sp³)-H bond at α site will dissociate. As the phonon modes of an adsorbed molecule can be excited by thermal treatment, we deduce that the cleavage of C(sp³)-H bonds at α sites should be at lower temperature than those at β sites..." - This is a questionable concept. The authors need to compare the C-H bond dissociation energies, i.e., the depths of the potential energy minima, and not the vibration frequencies, which correspond to the second derivatives at the potential minima. Even though there are sometimes limited correlations between these two quantities (i.e., dissociation energies and frequencies), it is difficult to understand why the vague argument with the frequencies is made here when the bond energies could be obtained from the same calculations. The calculated barriers shown in Figure 2 are much more useful in this context. Regarding the statement "the phonon modes on an adsorbed molecule can be excited by thermal treatment", the authors should also keep in mind that a thermal energy of 355 meV corresponds to a very high temperature of >4000 K. So, a thermal excitation of this vibration is certainly not the decisive factor for bond dissociation here, especially as multiple excitations would be necessary to induce thermal dissociation. What the authors observe here is first and foremost a surface-induced catalytic reaction and not a thermal cracking of C-H bonds due to excessive vibrational excitation.*

Response 4: We agree that there are limited correlations between dissociation energies and frequencies. Actually, what we that to say is that the frequency can be used to correlate to either the bond strength or the inter-atomic interaction (*Nat. Energy* 4, 408-415 (2019); *J. Am. Chem. Soc.* 122, 11660-11669 (2000); *J. Phys. Chem. A* 106, 9042-9052 (2002); *Polymer* 43, 6551-6559 (2002)). We are sorry for the confusion. Here, we only want to emphasize that the vibrational frequencies of C-H bond at α site and β site are equivalent in a freestanding molecule but become different after being adsorbed on Ag(100) substrate. The difference indicates that the bond strength of these C(sp^3)-H will be different.

We also agree that our observation is a surface-induced catalytic reaction and not a thermal cracking of C-H bonds due to excessive vibrational excitation. We deleted related descriptions and revised the related discussions in the manuscript (page 5 line 20) as follow:

"The phonon energies of C(sp^3)-H stretching mode can be used to correlate to the bond strength (*Nat. Energy* 4, 408-415 (2019); *J. Am. Chem. Soc.* 122, 11660-11669 (2000); *J Phys. Chem. A* 106, 9042-9052 (2002); *Polymer* 43, 6551-6559 (2002)), which can serve as a tentative prediction for the selective bond dissociation. It indicates the C(α)-H bond with a relatively low vibrational frequency is easier to dissociate than C(β)-H bond."

Comment 5: Why is the barrier for dissociation of the C-H bond H3 much higher (1.97 eV) when starting from the metastable state 2, compared to the barrier for dissociation of the same bond when starting from the initial state 1 (1.10 eV)?

Response 5: We carefully compared the atomic structures of initial states, transition states and final states in the two dehydrogenation pathways, as shown below in Figure R8. During two dehydrogenation processes, the adjacent benzene rings are both distorted (marked by the red dashed circles in Figure R8(a-b)), which decreases the planarity of conjugated molecules. We notice the out-of-plane buckling of **TS-3a** is larger than that of **TS-2a(1)**. The decrease of planarity of **TS-3a** leads higher barrier due to a less conjugated transition state.

Meanwhile, we analyzed the projected density of states (PDOS) onto the C(β) and adjacent N atoms in the initial states and transition states of the two processes. The results show both p_x and p_z orbitals contribute to the PDOS in C and N atoms in

transition state **TS-3a**, while only p_z orbital contributes to that in **TS-2a(1)**, as shown in Figure R8 (c-d), which further confirms the decrease of conjugation of **TS-3a**. Therefore, we conclude that the decrease of conjugation of **TS-3a** leads higher barrier in the pathway starting from metastable state **2** than that from the initial state **1**.

Figure R8 (Figure S33). (a, b) Atomic structures (top panel) and relative energies (bottom panel) of initial states, transition states and final states in two dehydrogenation processes involving H3 atoms. (c, d) Calculated PDOS onto the C(β) and adjacent N atoms in initial states **1**, **2** and transition states **TS-2a(1)**, **TS-3a**. The white, black and light blue balls represent Ag, carbon and nitrogen atoms, respectively. The hydrogen atoms bond with C(β) and C(α) are colored in red and blue, respectively. Other hydrogen atoms are colored in light pink.

We added Figure R8 as Figure S33 and relative discussions in the manuscript page 10 line 7 and Supplementary Information as follow:

In the manuscript: " We notice the barrier for dissociation of the C(sp^3)-H3 bond is much higher (1.97 eV) when starting from the metastable state **2**, compared to that starting from the initial state **1** (1.10 eV). The reason for the increased barrier is a less conjugated transition state **TS-3a** in the pathway starting from metastable state **2** (Figure S33)."

In the Supplementary Information: "During two dehydrogenation processes, the adjacent benzene rings are both distorted (marked by the red dashed circles in Figure S33(a-b)), which decreases the planarity of conjugated molecules. The out-of-plane buckling of **TS-3a** is larger than that of **TS-2a(1)**. The decrease of planarity of **TS-3a**

compared with **TS-2a(1)** leads to a higher barrier due to a less conjugated transition state. Meanwhile, we analyzed the PDOS onto the C(β) and adjacent N atoms in the initial states and transition states of the two processes. The results show both p_x and p_z orbitals contribute to the PDOS in C and N atoms in transition state **TS-3a**, while only p_z orbital contributes to that in **TS-2a(1)**, as shown in Figure S33 (c-d), which further confirms the decrease of conjugation of **TS-3a**. Therefore, we conclude that the decrease of conjugation of **TS-3a** leads to a higher barrier in the pathway starting from metastable state **2** than that from the initial state **1**."

***Comment 6:** Nanoribbons: The only large-scale SPM image (Figure 4a) shows just one ribbons with several repeat units resembling the zoom-in image in Figure 4b. All other products in the large-scale image show partially distorted shapes suggesting that they have different structures. The authors should provide statistical information regarding the yield of the W-shaped ribbons based on additional large-scale SPM data (which should be displayed in the SI).*

Response 6: We have added the statistical data on the yield of the W-shaped ribbons based on both large-scale and high-resolution STM images (Figures R9 and R10). As we discussed in the manuscript, **3** is prochiral on surface. When annealed to 673 K, the ribbons would consist of either homo-chiral or hetero-chiral units, among which only all hetero-chiral-coupled units can form uniform W-shaped ribbons. In principle, the possibilities are equal for homo-chiral or hetero-chiral coupling of monomers **3**. We analyzed 344 linkages in the ribbons in five high-resolution STM images and reveal that there are 47.7% hetero-chiral linkages and 52.3% other linkages (including homo-chiral linkages and unidentified linkages, shown in Figure R9). Due to the randomness of homo-chiral or hetero-chiral coupling, most ribbons are made of both hetero- and homo-chiral linkages, which results in distorted shapes. Even though, there are at least two W-shaped ribbons that consist of more than 4 monomers in every 100×100 nm² area, according to our large-area STM images (Figure R10).

Figure R9 (Figure S38). (a-e) STM images (20 nm × 20 nm) of ribbons acquired with a CO functionalized tip. The W-shape ribbon **5** is constructed by enantiomers with different chirality. We highlight two enantiomers with different chirality using blue and red shadows. (f) Statistics on the population of hetero-chiral linkages and other linkages.

Figure R10 (Figure S39). Large-scale (100 nm × 100 nm) STM images of the ribbons. The red boxes highlight the W-shaped ribbons that consist of more than 4 monomers.

We have added the following discussion in the manuscript (page 15 line 2).

"We analyzed 344 linkages in the ribbons in five high-resolution images, and found that 47.7% of them are hetero-chiral linkages (Figure S38). Due to the randomness of homo-chiral or hetero-chiral coupling, most ribbons are made of both hetero- and homo-chiral linkages, resulting in distorted shapes. Even though, there are at least two W-shaped ribbons that consist of more than 4 monomers in every 100 × 100 nm² area, according

to our large-area STM images (Figure S39)."

Response to Reviewer 3:

The manuscript by Gao et al. describes combined theoretical and experimental research on the hierarchical C-H bond activation in molecules synthesized from precursors directly on the Ag substrate. The investigation falls into the recently developing area of the on-surface synthesis approach. I believe that the main target of the manuscript is appealing for the broad audience. In general the research is clearly described. Before the manuscript can be accepted for publication there are, however, a few issues that the authors should clarify. I will briefly describe them below:

We thank the reviewer for his or her time and helpful comments. We appreciate the recognition that “*the main target of the manuscript is appealing for the broad audience*”.

Please find below the responses to the comments.

Comment 1: *the authors base their identification of the final structure on Ag (differentiation between 3 and 3') on the DFT calculations, i.e. the bond lengths and DOS. Have the authors tried to support the calculations by the experimental scanning tunneling spectroscopy measurements? The combination of theoretical and experimental reasoning would make the attribution of the actual structure much more reliable.*

Response 1: We have added STS on PAMY dimer (**3**), as shown by the red curve in Figure R11. The STS follows the calculated DOS spectral shape of **3** on Ag(100) (the light coral shadow in Figure R11) in combination with our extensive comparative studies in Figure 3 and AFM structural elucidation, which supports our conclusions for the structure assignment of **3**.

Figure R11. (Figure S36) Experimental dI/dV mapping on **3**. Red and gray lines are STS on **3** and on Ag(100), respectively. The inset is the STM image of a **3** on Ag(100). The scale bar is 2 nm. The red and grey dots indicate the positions we obtain the STS. The light coral shadow is calculated projected density of states (PDOS) of **3** on Ag(100). The STS follows the calculated DOS spectral shape of **3** on Ag(100) (the light coral shadow).

We have added the following sentence in the manuscript (line 18, page 12) and the Figure R11 as Figure S36 in the Supplementary Information.

In the manuscript: “Combined with the STS measurement which follows the calculated DOS spectral shape of **3** on Ag(100) (Figure S36), we assign the product as **3** with a planar configuration.”

Comment 2: in the description of the dehydrogenation path the energy differences for the two first steps between the lowest and second lowest possible paths are very small, in the first step the difference between the extraction of H1 and H3 is only 0.05 eV and in the second between the H2 and H3 is only 0.03 eV. However, in the sequential description the authors describe possible paths starting only from the lowest process in the previous step. For instance one could consider extraction of H3 in the first step and then the other process (which is not considered in the description). Although the first process would be higher in energy (H3 versus H1), the second one could be lower. I understand that the path described in Figure 2 is consistent with experimental findings, but my concern is whether the calculations describe the reality precisely enough, especially taking into account the relatively small differences in the two first steps.

Response 2: We thank the reviewer for suggesting this approach. We have now studied the dehydrogenation path of H3 in the first step and then the others. Two additional pathways are calculated: extraction of H3 first and then H4, extraction of H3 first and then H2, as shown in Figure R12. After the removal of H3 atom, the energy barriers for the removal of H4 and H2 atoms are 1.11 eV (H4), 1.07 eV (H2), respectively. The energy barriers are higher than the dehydrogenation sequence of H1 first (1.05 eV) and then H2 (0.57 eV). The new calculations strengthen our statement that the dehydrogenation sequence of H1, H2, H3 and H4 is favored.

Figure R12 (Figure S32). DFT calculations on dehydrogenation barriers of the two *ortho* C(sp^3) atoms in the N-heterocycles in **1 on Ag(100).** (a) Schematics of initial state (1), metastable state and final state of two dehydrogenation processes (H3 first and then H4, H3 first and then H2). (b) DFT-calculated energy profiles along the two dehydrogenation paths. The energy barriers are higher than the dehydrogenation sequence of H1 first and then H2, which strengthen our statement that the dehydrogenation sequence of H1, H2, H3 and H4 is favored.

The reviewer is concerned about *"the relatively small differences in the two first steps: in the first step the difference between the extraction of H1 and H3 is only 0.05 eV and in the second between the H2 and H3 is only 0.03 eV."* The energy barrier is related to the thickness of the substrate. In the previous calculations, three layers of Ag substrate are used. In experiments, the Ag substrate is rather thicker than that in our model. Thus, we increase the Ag substrate to five layers to test its effects on the energy barriers. The calculated energy barrier difference between the extraction of H1 and H3 is increased to 0.12 eV in the first step, as shown in Figure R13. Similarly, the energy barrier difference in the second step between the H2 and H3 is increased to 0.08 eV, which can describe the experimental observations. Though our calculated energy barriers may not

be the same as those in the experiments due to approximation, the relative differences do make sense.

Figure R13 (Figure S30). DFT calculations on the dehydrogenation barriers of *ortho* C(sp^3) atoms in the N-heterocycles for molecule 1 on five layers Ag(100) for the first two steps. The energy favourable profile is highlighted while the others are in light colours. According to the energy profile, the dehydrogenation sequence is H1, H2.

Moreover, in the previous Supplementary Information (S26), we provided *ab initio* molecular dynamics (AIMD) simulation of TPPIP on Ag(100) with three Ag layers. Owing to the timescale accessible to AIMD simulation, we can't carry out an AIMD simulation of TPPIP molecule on Ag(100) for ~1 h at 573 K to observe its dehydrogenation process. There is a widely used method to carry out MD simulation at high temperature with a short timescale to explain a process that happened at a low temperature with a long timescale. Hence, our AIMD simulation for TPPIP molecule on Ag(100) is performed at 1400 K for 3 ps, then 1600 K for 3ps and 2400 K for 1.5 ps. We took two snap shots as shown in Figure R14. According to the AIMD results, at 1400 K, H1 detached first, while the other H atoms still bond with C atoms (Figure R14a). The second H atom (H2) detached until the temperature increased to 2400 K, while the other H atoms, especially the H atoms at β sites, remain undetached (Figure R14b). This is in accordance with the sequence of dehydrogenation as predicted by the DFT-calculated energy profile.

Figure R14 (New Figure S31). Molecular dynamics (MD) simulation of TPPIP (1) at 1400 K for 3 ps, 1600 K for 3 ps, 2400 K for 1.5 ps. (a) Top and side views of the snap shot at 1400 K. The H at α site (H1, marked with red shadow) detached first, while the other H atoms still bond with C atoms. The H atoms at β sites are marked in light blue circles. (b) Top and side views of the snap shot at 2400 K. The H at the other α site (H2, marked with red shadow) detached, while the other H atoms still bond with C atoms. (c) Temperature variation with time, and the blue dots denote the times of snap shots taken in (a-b).

We have added the Figure R12-14 as Figure S30-S32 and the following discussion in the Supplementary Information.

"We find the calculated energy barrier difference between the removal of H1 and H3 in the first step and that of H2 and H3 in the second step are increased to 0.12 eV and 0.08 eV, respectively, when we consider more layers of substrate."

We also added the description of AIMD simulation in theoretical method and revised Figure R14 as new Figure S31.

In theoretical method (page 17 line 15): "For *ab initio* MD simulations, a canonical (NVT) ensemble was used at 1400 K, 1600 K and 2400 K. The high temperature is used to facilitate the transition process. The time interval between each step is 1 fs."

Comment 3: *the authors say that the DFT calculation predicts low band gap of the ribbons, have the authors attempt the STS measurements to extract the properties from the experiment?*

Response 3: Thank you for this suggestion, we have performed STS measurements on the ribbons on Ag(100) substrate. However, the states of the ribbons are highly hybridized with the substrate, as shown in Figure R15, whereby multiple features between -1 and 0.5 V are observed. Similarly, the electronic states of graphene nanoribbons zigzag-edged graphene nanoribbons have been shown to hybridize with the substrate (*Nature* 531, 489-492 (2016)).

Figure R15 (Figure S42). Experimental dI/dV mapping on the ribbon on Ag(100) substrate.

We have added Figure R15 as Figure S42 in the Supplementary Information.

Comment 4: *minor points: I strongly recommend that the authors read carefully the text as there are several minor language errors and sentences difficult to follow. For instance already the sentence "Experimentally depositing N-PH..." in the abstract may be misleading to suggest that the hierarchical dehydrogenation has been achieved through NC-AFM. In the manuscript there are more wierd sentences; the red ovals in Figure 3 are barely visible;*

Response 4: We thank the reviewer for the careful reading of our manuscript. We have

revised them following the suggestions:

1) The sentence in the abstract (page 2 line 7) is revised as "**Density functional theory calculations reveal that the** adsorption of N-PH on Ag(100) differentiates the activity of the four *ortho* C(sp^3) atoms in the N-heterocycles into two groups, suggesting a selective dehydrogenation, **which is demonstrated by sequential-annealing experiments of N-PH/Ag(100).**"

2) The red ovals in Figure 3 are highlighted as follows:

Figure 3. STM and nc-AFM images of 1 upon sequential annealing on Ag(100). (a-d) As-deposited 1 on the Ag(100). The long axis of a 1 monomer is marked with yellow dash line. (e-h) After annealing at 573 K for 30 mins, most 1 transformed to 2. (i-l) After annealing at 623 K for 30 mins, most of the molecules transformed to 3. (a, e, i) Large scale and (b, f, j) zoomed in STM images. (c, g, k) Zoomed in and (d, h, l) Laplacian filtered bond-resolved nc-AFM images, corresponding chemical structures are superposed in (d, h, l). The red dotted circles in the middle panels highlight the *ortho* C-H positions. Scanning parameters: the STM images: $V_s = -30$ mV, $I_t = 40$ pA; the nc-AFM images: Amplitude = 100 pm. Scale bars: (a, e, i) 5 nm, (b-d, f-h, j-l) 5 Å.

In summary, I find the manuscript interesting and appealing for the broad audience and believe that it could be accepted for publication once the above mentioned issues are clarified by the authors.

Response: We thank the reviewer for the constructive comments. We clarified the issues mentioned by the reviewer in a satisfactory manner. We believe the revised manuscript can be accepted for publication.

Response to Reviewer 4:

Comment 1: *The authors herein present the study of the stepwise thermal dehydrogenation of a polycyclic heteroaromatic hydrocarbon on a Ag (100) surface. At the center of this work lies the observation of a hierarchical temperature-controlled dehydrogenation that can be correlated with the subtle differences in the C-H BDE or vibrational ZPE of individual C-H bonds upon adsorption on the surface. The authors highlight this work as an extension of the field of C-H activation. This is clearly done in an effort to raise the impact of their work but appears as a significant stretch. C-H activation usually implies the chemical substitution of C-H bond by a higher value C-X bond not just the cleavage of a hydrogen atom. The more appropriate analogy may be found in the conventional radical mediated oxidation reactions.*

Response 1: We agree with the reviewer that "C-H activation usually implies the chemical substitution of C-H bond by a higher value C-X bond". We want to emphasize that the C-H bond cleavage is a key process in C-H activation. Actually, the C-H activation in our work results in a conversion of sp^3 -C to sp^2 -C on the metal surface, rather than the formation of a higher value C-X bond, which belongs to intramolecular C-H activation. Correspondingly, we realize C-N transforming from C(sp^3)-N single bond to C(sp^2)-N ionic bond and C-C transforming from C(sp^3)-C bond to C(sp^2)-C bond through *ortho* C-H activation. In addition, the previous study reported similar bond order reorganization via C-H activation (*J. Phys. Chem. Lett.* 11, 9850-9855 (2020)); *Nature Communications* 9, 1198 (2018)). In order to make our work clear, we revised our title as "Selective C-H bond-activation **induced C-C/C-N bond-transformation** on surface and enabled formation of N-doped graphene nanoribbons".

In this regard, we politely disagree that our work is an analogy for "conventional radical mediated oxidation reactions". To the best of our knowledge, the reaction sites in conventional radical mediated oxidation reactions are usually the radical sites [*Chem. Commun.* 56, 6907 (2020); *Nature Reviews Chemistry* 6, 405 (2022)]. In our work, the precursor **1** is not a radical, as shown in Figure R16. There is a radical site (C(sp^2) at α site) after the first H atom (H1, the C(sp^3) at α site) detached (**2a**). However, the further dehydrogenation didn't happen at the radical site. Moreover, after the first two H atoms (H1 and H2) were detached, precursor **1** transformed into an intermediate (**2**) with a quinodimethane core rather than a radical form.

Figure R16. Schematics of initial state (**1**), three metastable states (**2a**, **2**, **3a**) and final state (**3**) of dehydrogenation process of the four *ortho* C(sp^3) atoms in the N-heterocycles in **1** on Ag(100).

Comment 2: Barred these comments the authors should clarify the key advancements to the field and timeliness that would justify publication in *Nature Communications*.

Response 2: Previous works of selective activation of C-H bonds lacks selectivity among equivalent/quasi-equivalent *ortho* C atoms in heterocycles (*Acc. Chem. Res.* 45, 788-802, (2012); *Acc. Chem. Res.* 45, 936-946, (2012); *Chem. Soc. Rev.* 43, 6906-6919, (2014); *Chem. Rev.* 117, 9302-9332, (2017)). In this regard, the step-wise, kinetic activation of equivalent/quasi-equivalent of *ortho* C(sp^3) atoms should be a major achievement in the field. Here, we reported the realization of surface-induced selective dehydrogenation of quasi-equivalent *ortho* C(sp^3) atoms in a N-heterocycle for the first time. Moreover, this is the first example to realize the synthesis of graphene nanoribbons by using the C-H bond activation chemistry, rather than the classical Ullmann-type coupling reaction for the polymerization together with the intramolecular cyclodehydrogenation. We believe this work will be regarded as a pivotal advance by experts in organic chemistry and surface science, and is sufficiently important to be published in *Nature Communications*.

We added the following sentences in the introduction and abstract in the manuscript to emphasize the key advancement to the field.

In the introduction (page 3 line 2): "Carbon-hydrogen (C-H) bond activation in a molecule is an efficient yet challenging route towards synthesis of complex organic compounds due to the existence of multiple reactive sites. By introducing heteroatoms, such as nitrogen and sulfur atoms into the organic skeleton, the *ortho* C atoms can be more reactive than other C atoms. This strategy, however, lacks selectivity among equivalent/quasi-equivalent *ortho* C atoms, as previous studies show all the *ortho* C

atoms have almost equal probability to be activated. Therefore, in route towards ultimate control over C-H selectivity, the selective activation of equivalent/quasi-equivalent *ortho* C(sp^3)-H with new methodologies would be a major achievement in the field."

In the abstract (page 2 line 10): "Further annealing leads to the formation of N-doped graphene nanoribbons with partial corannulene motifs, realized by the C-H bond activation process."

Comment 3: *It remains unclear how the observed selectivity could be used as a design strategy barred the not insignificant computational effort that was necessary to disentangle the proposed mechanisms.*

Response 3: We appreciate the reviewer's suggestion that our conclusions should be rephrased to help advance knowledge in the field. According to our prediction of differentiating the activity in quasi-equivalent sites through adsorption on substrate, the design strategy can be selecting suitable substrate on which the adsorption of the precursors will differentiate the quasi-equivalent sites, enabling the selective C-H activation and further specific bond transformation or chemical substitution for new reactivity paradigms.

Therefore, we now added the following discussion in the conclusion in the manuscript (page 16 line 9).

"Our findings provide a design strategy to differentiate the activity in quasi-equivalent sites with *ortho* C(sp^3) atom in a N-heterocycle by choosing suitable substrates to realize selective dehydrogenation. This strategy enables the selective C-H activation and further specific bond transformation or chemical substitution for new reactivity paradigms."

Comment 4: *The reliable calculation of the preferred adsorption geometry of an arbitrary molecule barred any experimental data is an enormous challenge in itself.*

Response 4: We agree with the reviewer that global structure optimization is a major field of research, and are happy that this is recognized as a major effort. Our groups have been dedicating to the specific problem of global optimization on substrates for almost a decade, on graphite (*J. Phys. Chem. Lett.* 2010, 1, 23, 3407–3412), on metals

(*Phys. Rev. Lett.* 2006, 97, 156105; *Phys. Rev. Lett.* 2007, 99, 106402; *Phys. Rev. Lett.* 2010, 104, 166101; *J. Am. Chem. Soc.* 2020, 142, 24, 10673-10680; *Nature Communications* 11, 1490 (2020); *Nature Communications* 10, 3599 (2019); *Nature Communications* 9, 3277 (2018)). In these works, the calculated most stable configurations agree well with the experimental observations. Therefore, the calculation of the preferred adsorption geometry of an arbitrary molecule should be reliable.

In this work, we performed potential energy surface (PES) scan at different rotation angles. The PES scan at different rotation angles was plotted, as shown in Figure R17, which helps to find the most stable adsorption structure at the energy minimum. We took 45 possible configurations with each rotation angle differs by 2 degree. In each configuration, the top two layers of substrate were fully relaxed, while the molecule was relaxed in z direction with a fixed in-plane orientation. Then PES is acquired after performing single point energy calculation at each rotation angle. The calculated PES shows that the configurations with rotation angles of 22 and 24 degree exhibit the lowest energies with a difference of 2 meV, as shown in Figure R6b. After further relaxation of the configuration with a rotation angle of 24 degree, it relaxes to the configuration with a rotation angle of 22 degree. Therefore, we exclude that there are even lower energies at other angles.

Figure R17 (Figure S25). DFT Calculated potential energy surface at different rotation angles. (a) Schematics of adsorption configuration of **1** on Ag(100) with different orientations. The long axis of **1**, the [110] direction of Ag(100), and the rotation axis are marked with black dash line, blue dash arrow and blue line. The rotation angle is denoted by θ . (b) DFT calculated potential energy surface at different rotation angles θ . We took 45 possible configurations with each rotation angle differs by 2 degree. In each configuration, the molecule is fully relaxed in z

direction. The configurations with rotation angles of 22 and 24 degree exhibit the lowest energies with a difference of 2 meV. Further relaxation finds that the configuration with a rotation angle of 24 degree changes to that with a rotation angle of 22 degree, which is the most possible configuration of molecule **1** on Ag(100).

We have added Figure R17 as Figure S25 in Supplementary Information and related discussion in the manuscript (page 5 line 12).

"We first performed potential energy surface (PES) calculations by scanning 45 configurations with different rotation angles, as shown in Figure S25. Further relaxation finds that the configuration with rotation angle $\theta = 22^\circ$ (light coral stripe in Figure 1b) is most stable for **1** on Ag(100)."

***Comment 5:** Adding on top of that truly subtle differences in the ZPE of 6 meV that apparently gives rise to the observed selectivity is far from a practical predictive model and well within the error of DFT calculations.*

Response 5: It is worth mentioning that what we calculated is the specific C(sp³)-H bond vibration frequencies but not zero-point energy. Zero-point energy refers to the vibration energy of a system at absolute zero, which relates to all vibration modes in a system. The analysis of C(sp³)-H bond vibration frequencies at α and β sites is to compare the bond strength of specific C-H bonds here.

Moreover, we also provide more calculation results to reveal the selective activation of C-H bonds, for example, dehydrogenation energy barriers of the four C(sp³)-H bonds at α and β sites using the climbing-image nudged elastic band (CI-NEB) method. The preference of the dehydrogenation path can be revealed by comparing the relative dehydrogenation energy barriers. This is a commonly used method in surface-science community (*Science* 334, 213-216 (2011); *J. Am. Chem. Soc.*, 141, 3550 (2019)).

Beyond the general comments above the reviewers are left with two key questions that need to be resolved before this manuscript could be reconsidered.

***Comment 6:** The authors introduce RAC(alpha)/C(beta) as a the "relative activity" of C(sp³)-H bonds flanking the N-atoms. The validity and the origin of this reference has to be discussed in the manuscript. The explanation provided on page 5 is more than just unclear and leaves the reader with an incomplete understanding of the context of this comparison.*

Response 6: We thank the reviewer for suggesting that the arguments behind our

conclusion should be clarified. Our claim of "close relationship between dissociation of the C-H bond and HOMO states" is based on the frontier molecular orbital (FMO) theory (*J. Chem. Phys.* 20, 722-725 (1952)), which states that the frontier orbitals, i.e. the HOMO and the LUMO, are mainly responsible for chemical reactions. A recent work (*Communications Chemistry* 2, 31 (2019)) shows that the occupied C-H orbital would be expected to interact with an unoccupied *d* orbital of Ni to activate the corresponding C-H bond, in agreement with Fukui's FMO theory. Moreover, a recent work uses the differences in the electron density of singly occupied molecular orbital at the reactive position to compare the reactivity difference. (*Nature Communications* 10, 2706 (2019)).

Since the *ortho* C(*sp*³) atoms (C(α) and C(β)) in N-heterocycles are active sites, their contribution to the HOMO or LUMO will determine their activity. The dissociation of C-H bond will involve HOMO due to the loss of H atom. Therefore, in order to reveal the relationship between the dissociation of C-H bond and the frontier orbitals, we investigate the contribution of these *ortho* C(*sp*³) atoms to the HOMO state through DOS on these atoms (PDOS). Due to the split HOMO peaks after the molecule is adsorbed on substrate, we calculate the contribution of C(α) or C(β) by integrating the PDOS from the original HOMO to the Fermi level as shown in Figure R18 marked with a light coral background. The $RA_{C(\alpha)/C(\beta)}$ is then defined as the ratio of contribution from C(α) to the HOMO and that from C(β). For $RA_{C(\alpha)/C(\beta)}$ larger than 1, the C(α) is more active than C(β), and vice versa.

Figure R18 (Figure S26). (a) Projected density of states (PDOS) on C atoms at α site (upper panel) and β site (lower panel) of TPPIP on Ag(100). (b) The schematic of molecule 1 on Ag(100).

We have now added the following discussion in the manuscript (page 5 line 24) to explain the close relationship between dissociation of C-H and HOMO.

"To further solidify the relative activity of C(sp^3) at α and β sites, we analyze their contribution to the frontier orbitals. Based on the frontier molecular orbital theory (*J. Chem. Phys.* 20, 722-725 (1952)), the highest occupied molecular orbital (HOMO) and the lowest unoccupied molecular orbital (LUMO), are mainly responsible for chemical reactions (*Communications Chemistry* 2, 31 (2019); *Nature Communications* 10, 2706 (2019)). Since the *ortho* C(sp^3) atoms (C(α) and C(β)) in N-heterocycles are active sites, their contributions to the HOMO or LUMO will determine their activity. The dissociation of C-H bond involves HOMO due to the loss of H atom. Therefore, we defined a term $RA_{C(\alpha)/C(\beta)}$ as the ratio of electron density of HOMO contributed from C(α) to that from C(β) (detailed information can be found in Supplementary Information S26) and plotted the data by the purple stars in Figure 1b. "

We added Figure R18 as Figure S26 and description of $RA_{C(\alpha)/C(\beta)}$ in the Supplementary information.

"The $RA_{C(\alpha)/C(\beta)}$ is defined as the ratio of electron density of HOMO contributed from C(α) to that from C(β). Due to the split HOMO peaks after the molecule is adsorbed on substrate (Figure S26 (a)), we obtain the contribution of C(α) or C(β) by integrating the PDOS from the original HOMO to the Fermi level as shown in Figure S26 marked with a light coral background."

Comment 7: *Based on the extended data provided as part of the supporting information it appears like the C-H bond broken as part of the dehydrogenation steps is actually the C-H bond pointing towards the surface. The authors however focus almost exclusively on the discussion of the ZPE of the C-H bonds pointing away from the surface (see Figure 1). This is bizarre as the discussed C-H bonds remain intact in the product and are therefore "irrelevant" (beyond a secondary effect) to the selectivity. Should the discussion not focus on the actual C-H bond that is broken or are the authors suggestion a mechanism where the H-atom is removed without assistance from the underlying Ag(100) substrate? There is plenty of evidence in the literature that suggest that all dehydrogenations on coinage metal surfaces rely on the transfer of a H-atom to the surface (geometrically only possible if the C-H bond points into the substrate).*

Response 7: We thank the reviewer for the careful reading of our manuscript and agree with the reviewer that the dehydrogenation process is assisted with the underlying Ag(100) substrate. In the manuscript, what we discussed was the C-H bond pointing towards the surface, including the bond vibration analysis in Figure 1c. However, the

illustration of C-H bond in Figure 1c and the lower panel in Figure 2 are misleading, therefore, we revised the bond shape of the four *ortho* C-H bonds, as shown in Figure R20 (New Figure 1c). We have also added the original data of the calculated C-H vibration modes, as shown in Figure R19.

Figure R19 (Figure S27). DFT calculations of the specific C-H vibrational modes of TPPIP on Ag(100). (a) Two degenerate phonon modes at 355 meV. The two modes represent the C-H stretch at α sites. (b) Two degenerate phonon modes at 361 meV. The two modes represent the C-H stretch at β sites.

Figure R20 (Figure 1c) Specific $C(sp^3)$ -H vibrational modes in the most stable configuration. Left panel: two degenerate phonon modes at 355 meV represent the $C(sp^3)$ -H stretch at α sites (marked in blue). Two degenerate phonon modes at 361 meV represent the $C(sp^3)$ -H stretch at

β sites (marked in red). Middle and right panels: possible intermediate **2** and product PAMY dimer after detachment of H atoms from C atoms at α and β sites.

Figure R21 (The lower panel in Figure 2). DFT calculations on dehydrogenation barriers of the four ortho C(sp³) atoms in the N-heterocycles in **1** on Ag(100). The lower panel, schematics of initial state (**1**), three metastable states (**2a**, **2**, **3a**) and final state (**3**) of dehydrogenation process we used in the calculations.

Minor corrections:

Page 3 “...surface characterizing...” change to “...surface characterization...”

Page 6 “...density of LUMO distributes at all...” change to “... density of LUMO is distributed over all...”

Page 6 “...inclines...” change to “...prefers...”

Response: We thank the reviewer for the careful reading of our manuscript. We have corrected all the typos.

REVIEWERS' COMMENTS

Reviewer #1 (Remarks to the Author):

I confirmed the author's responses to my comments. Overall, they addressed my comments properly. Therefore, in my evaluation, it is now ready for acceptance.

Reviewer #2 (Remarks to the Author):

The authors have provided a careful and comprehensive revision of their work and have adequately addressed all issues raised by the reviewers. Publication of this manuscript in Nature Communications is recommended.

Reviewer #3 (Remarks to the Author):

Based on the careful reading of the revised manuscript it appears clear to me that the authors have undertaken a major effort to improve the quality of the manuscript. I believe that the authors have included important modifications to answer the raised questions. In my opinion the current version of the manuscript reporting the selective C-H bond activation on Ag meets the criteria for publications in Nature Communications and therefore I recommended the revised manuscript for publication.

Reviewer #4 (Remarks to the Author):

The authors have made a significant effort to revise the manuscript. This additional work has led to a significant improvement. The current manuscript should be considered for publication in Nature Communications.